# LoCoDL: Communication-Efficient Distributed Learning with Local Training and Compression

**Laurent Condat, Artavazd Maranjyan & Peter Richtárik**
Computer Science Program, CEMSE Division
King Abdullah University of Science and Technology (KAUST)
Thuwal, 23955-6900, Kingdom of Saudi Arabia
& SDAIA-KAUST Center of Excellence in Data Science and Artificial Intelligence
(SDAIA-KAUST AI)
`first.last@kaust.edu.sa`

## Abstract

In Distributed optimization and Learning, and even more in the modern framework of federated learning, communication, which is slow and costly, is critical. We introduce LoCoDL, a communication-efficient algorithm that leverages the two popular and effective techniques of Local training, which reduces the communication frequency, and Compression, in which short bitstreams are sent instead of full-dimensional vectors of floats. LoCoDL works with a large class of unbiased compressors that includes widely-used sparsification and quantization methods. LoCoDL provably benefits from local training and compression and enjoys a doubly-accelerated communication complexity, with respect to the condition number of the functions and the model dimension, in the general heterogenous regime with strongly convex functions. This is confirmed in practice, with LoCoDL outperforming existing algorithms.

## 1 Introduction

Performing distributed computations is now pervasive in all areas of science. Notably, Federated Learning (FL) consists in training machine learning models in a distributed and collaborative way (Konečný et al., 2016a;b; McMahan et al., 2017; Bonawitz et al., 2017). The key idea in this rapidly growing field is to exploit the wealth of information stored on distant devices, such as mobile phones or hospital workstations. The many challenges to face in FL include data privacy and robustness to adversarial attacks, but communication-efficiency is likely to be the most critical (Kairouz et al., 2021; Li et al., 2020a; Wang et al., 2021). Indeed, in contrast to the centralized setting in a datacenter, in FL the clients perform parallel computations but also communicate back and forth with a distant orchestrating server. Communication typically takes place over the internet or cell phone network, and can be slow, costly, and unreliable. It is the main bottleneck that currently prevents large-scale deployment of FL in mass-market applications.

Two strategies to reduce the communication burden have been popularized by the pressing needs of FL: 1) **Local Training (LT)**, which consists in reducing the communication frequency. That is, instead of communicating the output of every computation step involving a (stochastic) gradient call, several such steps are performed between successive communication rounds. 2) **Communication Compression (CC)**, in which compressed information is sent instead of full-dimensional vectors. We review the literature of LT and CC in Section 1.2.

We propose a new randomized algorithm named LoCoDL, which features LT and unbiased CC for communication-efficient FL and distributed optimization. It is variance-reduced (Hanzely & Richtárik, 2019; Gorbunov et al., 2020a; Gower et al., 2020), so that it converges to an exact solution. It provably benefits from the two mechanisms of LT and CC: the communication complexity is doubly accelerated, with a better dependency on the condition number of the functions and on the dimension of the model.

## 1.1 PROBLEM AND MOTIVATION

We study distributed optimization problems of the form

$$\min_{x \in \mathbb{R}^d} \ \frac{1}{n} \sum_{i=1}^{n} f_i(x) + g(x), \tag{1}$$

where $d \geq 1$ is the model dimension and the functions $f_i : \mathbb{R}^d \to \mathbb{R}$ and $g : \mathbb{R}^d \to \mathbb{R}$ are smooth. We consider the server-client model in which $n \geq 1$ clients do computations in parallel and communicate back and forth with a server. The private function $f_i$ is owned by and stored on client $i \in [n] := \{1, \ldots, n\}$. Problem (1) models empirical risk minimization, of utmost importance in machine learning (Sra et al., 2011; Shalev-Shwartz & Ben-David, 2014). More generally, minimizing a sum of functions appears in virtually all areas of science and engineering. Our goal is to solve Problem (1) in a communication-efficient way, in the general **heterogeneous** setting in which the functions $f_i$, as well as $g$, can be *arbitrarily different*: we do not make any assumption on their similarity whatsoever.

We consider in this work the strongly convex setting. That is, the following holds:

**Assumption 1.1** (strongly convex functions). The functions $f_i$ and $g$ are all $L$-smooth and $\mu$-strongly convex, for some $0 < \mu \leq L$.[1] Then we denote by $x^\star$ the solution of the strongly convex problem (1), which exists and is unique. We define the condition number $\kappa := \frac{L}{\mu}$.

Problem (1) can be viewed as the minimization of the average of the $n$ functions $(f_i + g)$, which can be performed using calls to $\nabla(f_i + g) = \nabla f_i + \nabla g$. We do not use this straightforward interpretation. Instead, let us illustrate the interest of having the **additional function** $g$ in (1), using 4 different viewpoints. We stress that we can handle the case $g = 0$, as discussed in Section 3.1.

● Viewpoint 1: *regularization*. The function $g$ can be a regularizer. For instance, if the functions $f_i$ are convex, adding $g = \frac{\mu}{2}\|\cdot\|^2$ for a small $\mu > 0$ makes the problem $\mu$-strongly convex.

● Viewpoint 2: *shared dataset*. The function $g$ can model the cost of a common dataset, or a piece thereof, that is known to all clients.

● Viewpoint 3: *server-aided training*. The function $g$ can model the cost of a core dataset, known only to the server, which makes calls to $\nabla g$. This setting has been investigated in several works, with the idea that using a small auxiliary dataset representative of the global data distribution, the server can correct for the deviation induced by partial participation (Zhao et al., 2018; Yang et al., 2021; 2024). We do not focus on this setting, because we deal with the general heterogeneous setting in which $g$ and the $f_i$ are not meant to be similar in any sense, and in our work $g$ is handled by the clients, not by the server.

● Viewpoint 4: *a new mathematical and algorithmic principle*. This is the idea that led to the construction of LoCoDL, and we detail it in Section 2.1.

In LoCoDL, the clients make all gradient calls; that is, Client $i$ makes calls to $\nabla f_i$ and $\nabla g$.

## 1.2 STATE OF THE ART

We review the latest developments on communication-efficient algorithms for distributed learning, making use of LT, CC, or both. Before that, we note that we should distinguish uplink, or clients-to-server, from downlink, or server-to-clients, communication. Uplink is usually slower than downlink communication, since the clients uploading *different* messages in parallel to the server is slower than the clients downloading *the same* message in parallel from the server. This can be due to cache memory and aggregation speed constraints of the server, as well as asymmetry of the service provider's systems or protocols used on the internet or cell phone network. In this work, we focus on the **uplink communication complexity**, which is often the bottleneck in practice. Indeed, the goal is to exploit parallelism to obtain better performance when $n$ increases. Precisely, with LoCoDL, the uplink communication complexity decreases from $\mathcal{O}\left(d\sqrt{\kappa}\log \epsilon^{-1}\right)$ when $n$ is small to

---

[1]A differentiable function $f : \mathbb{R}^d \to \mathbb{R}$ is said to be $L$-smooth if $\nabla f$ is $L$-Lipschitz continuous; that is, for every $x \in \mathbb{R}^d$ and $y \in \mathbb{R}^d$, $\|\nabla f(x) - \nabla f(y)\| \leq L\|x - y\|$ (the norm is the Euclidean norm throughout the paper). $f$ is said to be $\mu$-strongly convex if $f - \frac{\mu}{2}\|\cdot\|^2$ is convex.

$\mathcal{O}\left(\sqrt{d}\sqrt{\kappa}\log\epsilon^{-1}\right)$ when $n$ is large, where the condition number $\kappa$ is defined in Assumption 1.1, see Corollary 3.2. Many works have considered bidirectional compression, which consists in compressing the messages sent both ways (Gorbunov et al., 2020b; Philippenko & Dieuleveut, 2020; Liu et al., 2020; Philippenko & Dieuleveut, 2021; Condat & Richtárik, 2022; Gruntkowska et al., 2023; Tyurin & Richtárik, 2023b) but to the best of our knowledge, this has no impact on the downlink complexity, which cannot be reduced further than $\mathcal{O}\left(d\sqrt{\kappa}\log\epsilon^{-1}\right)$, just because there is no parallelism to exploit in this direction. Thus, we focus our analysis on theoretical and algorithmic techniques to reduce the uplink communication complexity, which we call communication complexity in short, and we ignore downlink communication.

**Communication Compression (CC)** consists in applying some lossy scheme that compresses vectors into messages of small bit size, which are communicated. For instance, the well-known rand-$k$ compressor selects $k$ coordinates of the vector uniformly at random, for some $k \in [d] := \{1, \ldots, d\}$. $k$ can be as small as 1, in which case the compression factor is $d$, which can be huge. Some compressors, such as rand-$k$, are unbiased, whereas others are biased; we refer to Beznosikov et al. (2020); Albasyoni et al. (2020); Horváth et al. (2022); Condat et al. (2022b) for several examples and a discussion of their properties. The introduction of DIANA by Mishchenko et al. (2019) was a major milestone, as this algorithm converges linearly with the large class of unbiased compressors defined in Section 1.3 and also considered in LoCoDL. The communication complexity $\mathcal{O}\left(d\kappa\log\epsilon^{-1}\right)$ of the basic Gradient Descent (GD) algorithm is reduced with DIANA to $\mathcal{O}\left((\kappa + d)\log\epsilon^{-1}\right)$ when $n$ is large, see Table 2. DIANA was later extended in several ways (Horváth et al., 2022; Gorbunov et al., 2020a; Condat & Richtárik, 2022). An accelerated version of DIANA called ADIANA based on Nesterov Accelerated GD has been proposed (Li et al., 2020b) and further analyzed in He et al. (2023); it has the state-of-the-art theoretical complexity.

Algorithms converging linearly with biased compressors have also been proposed, such as EF21 (Richtárik et al., 2021; Fatkhullin et al., 2021; Condat et al., 2022b), but the acceleration potential is less understood than with unbiased compressors. Algorithms with CC such as MARINA (Gorbunov et al., 2021) and DASHA (Tyurin & Richtárik, 2023a) have been proposed for nonconvex optimization, but their analysis requires a different approach and there is a gap in the achievable performance: their complexity depends on $\frac{\omega\kappa}{\sqrt{n}}$ instead of $\frac{\omega\kappa}{n}$ with DIANA, where $\omega$ characterizes the compression error variance, see (2). Therefore, we focus on the convex setting and leave the nonconvex study for future work.

**Local Training (LT)** is a simple but remarkably efficient idea: the clients perform multiple Gradient Descent (GD) steps, instead of only one, between successive communication rounds. The intuition behind is that this leads to the communication of richer information, so that the number of communication rounds to reach a given accuracy is reduced. We refer to Mishchenko et al. (2022) for a comprehensive review of LT-based algorithms, which include the popular FedAvg and Scaffold algorithms of McMahan et al. (2017) and Karimireddy et al. (2020), respectively. Mishchenko et al. (2022) made a breakthrough by proposing Scaffnew, the first LT-based variance-reduced algorithm that not only converges linearly to the exact solution in the strongly convex setting, but does so with accelerated communication complexity $\mathcal{O}(d\sqrt{\kappa}\log\epsilon^{-1})$. In Scaffnew, communication can occur randomly after every iteration, but occurs only with a small probability $p$. Thus, there are in average $p^{-1}$ local steps between successive communication rounds. The optimal dependency on $\sqrt{\kappa}$ (Scaman et al., 2019) is obtained with $p = 1/\sqrt{\kappa}$. LoCoDL has the same probabilistic LT mechanism as Scaffnew but does not revert to it when compression is disabled, because of the additional function $g$ and tracking variables $y$ and $v$. A different approach to LT was developed by Sadiev et al. (2022a) with the APDA-Inexact algorithm, and generalized to handle partial participation by Grudzień et al. (2023) with the 5GCS algorithm: in both algorithms, the local GD steps form an inner loop in order to compute a proximity operator inexactly.

**Combining LT and CC** while retaining their benefits is very challenging. In our strongly convex and heterogeneous setting, the methods Qsparse-local-SGD (Basu et al., 2020) and FedPAQ (Reisizadeh et al., 2020) do not converge linearly. FedCOMGATE features LT + CC and converges linearly (Haddadpour et al., 2021), but its complexity $\mathcal{O}(d\kappa\log\epsilon^{-1})$ does not show any acceleration. We can mention that random reshuffling, a technique that can be seen as a type of LT, has been combined with CC in Sadiev et al. (2022b); Malinovsky & Richtárik (2022). Recently, Condat et al. (2022a) managed to design a specific compression technique compatible with the LT mechanism of Scaffnew, leading to CompressedScaffnew, the first LT + CC algorithm exhibiting a doubly-accelerated complexity,

namely $\mathcal{O}\big(\big(\sqrt{d}\sqrt{\kappa}+\frac{d\sqrt{\kappa}}{\sqrt{n}}+d\big)\log\epsilon^{-1}\big)$, as reported in Table 2. However, CompressedScaffnew uses a specific linear compression scheme that requires shared randomness; that is, all clients have to agree on a random permutation of the columns of the global compression pattern. No other compressor can be used, which notably rules out any type of quantization.

## 1.3 A General Class of Unbiased Random Compressors

For every $\omega \geq 0$, we define $\mathbb{U}(\omega)$ as the set of random compression operators $\mathcal{C} : \mathbb{R}^d \to \mathbb{R}^d$ that are unbiased, i.e. $\mathbb{E}[\mathcal{C}(x)] = x$, and satisfy, for every $x \in \mathbb{R}^d$,

$$\mathbb{E}\Big[\big\|\mathcal{C}(x) - x\big\|^2\Big] \leq \omega\,\|x\|^2. \tag{2}$$

In addition, given a collection $(\mathcal{C}_i)_{i=1}^n$ of compression operators in $\mathbb{U}(\omega)$ for some $\omega \geq 0$, in order to characterize their joint variance, we introduce the constant $\omega_{\mathrm{av}} \geq 0$ such that, for every $x_i \in \mathbb{R}^d$, $i \in [n]$, we have

$$\mathbb{E}\left[\left\|\frac{1}{n}\sum_{i=1}^n \big(\mathcal{C}_i(x_i) - x_i\big)\right\|^2\right] \leq \frac{\omega_{\mathrm{av}}}{n}\sum_{i=1}^n \|x_i\|^2. \tag{3}$$

The inequality (3) is not an additional assumption: it is satisfied with $\omega_{\mathrm{av}} = \omega$ by convexity of the squared norm. But the convergence rate will depend on $\omega_{\mathrm{av}}$, which is typically much smaller than $\omega$. In particular, if the compressors $\mathcal{C}_i$ are mutually independent, the variance of their sum is the sum of their variances, and (3) is satisfied with $\omega_{\mathrm{av}} = \frac{\omega}{n}$.

## 1.4 Challenge and Contributions

This work addresses the following question: *Can we combine LT and CC with any compressors in the generic class $\mathbb{U}(\omega)$ defined in the previous section, and fully benefit from both techniques by obtaining a doubly-accelerated communication complexity?*

We answer this question in the affirmative. LoCoDL has the same probabilistic LT mechanism as Scaffnew and features CC with compressors in $\mathbb{U}(\omega)$ with arbitrarily large $\omega \geq 0$, with proved linear convergence under Assumption 1.1, without further requirements. By choosing the communication probability and the variance $\omega$ appropriately, double acceleration is obtained. Thus, LoCoDL achieves the same theoretical complexity as CompressedScaffnew, but allows for a large class of compressors instead of the cumbersome permutation-based compressor of the latter. In particular, with compressors performing sparsification and quantization, LoCoDL outperforms existing algorithms, as we show by experiments in Section 4. This is remarkable, since ADIANA, based on Nesterov acceleration and not LT, has an even better theoretical complexity when $n$ is larger than $d$, see Table 2, but this is not reflected in practice: ADIANA is clearly behind LoCoDL in our experiments. Thus, our experiments indicate that LoCoDL sets new standards in terms of communication efficiency.

## 2 Proposed Algorithm LoCoDL

### 2.1 Principle: Double Lifting of the Problem to a Consensus Problem

In LoCoDL, every client stores and updates *two* local model estimates. They will all converge to the same solution $x^\star$ of (1). This construction comes from two ideas.

**Local steps with local models.** In algorithms making use of LT, such as FedAvg, Scaffold and Scaffnew, the clients store and update local model estimates $x_i$. When communication occurs, an estimate of their average is formed by the server and broadcast to all clients. They all resume their computations with this new model estimate.

**Compressing the difference between two estimates.** To implement CC, a powerful idea is to compress not the vectors themselves, but *difference vectors* that converge to zero. This way, the algorithm is variance-reduced; that is, the compression error vanishes at convergence. The technique of compressing the difference between a gradient vector and a control variate is at the core of algorithms such as DIANA and EF21. Here, we want to compress differences between model estimates, not gradient estimates. That is, we want Client $i$ to compress the difference between $x_i$ and

Table 1: Communication complexity in number of communication rounds to reach $\epsilon$-accuracy for linearly-converging algorithms allowing for CC with independent compressors in $\mathbb{U}(\omega)$ for any $\omega \geq 0$. Since the compressors are independent, $\omega_{\text{av}} = \frac{\omega}{n}$. We provide the leading asymptotic factor and ignore log factors such as $\log \epsilon^{-1}$. The state of the art is highlighted in green.

| Algorithm | Com. complexity in # rounds | case $\omega = \mathcal{O}(n)$ | case $\omega = \Theta(n)$ |
|---|---|---|---|
| DIANA | $(1 + \frac{\omega}{n})\kappa + \omega$ | $\kappa + \omega$ | $\kappa + \omega$ |
| EF21 | $(1 + \omega)\kappa$ | $(1 + \omega)\kappa$ | $(1 + \omega)\kappa$ |
| 5GCS-CC | $\left(1 + \sqrt{\omega} + \frac{\omega}{\sqrt{n}}\right)\sqrt{\kappa} + \omega$ | $(1 + \sqrt{\omega})\sqrt{\kappa} + \omega$ | $(1 + \sqrt{\omega})\sqrt{\kappa} + \omega$ |
| ADIANA[1] | $\left(1 + \frac{\omega^{3/4}}{n^{1/4}} + \frac{\omega}{\sqrt{n}}\right)\sqrt{\kappa} + \omega$ | $\left(1 + \frac{\omega^{3/4}}{n^{1/4}}\right)\sqrt{\kappa} + \omega$ | $(1 + \sqrt{\omega})\sqrt{\kappa} + \omega$ |
| ADIANA[2] | $\left(1 + \frac{\omega}{\sqrt{n}}\right)\sqrt{\kappa} + \omega$ | $\left(1 + \frac{\omega}{\sqrt{n}}\right)\sqrt{\kappa} + \omega$ | $(1 + \sqrt{\omega})\sqrt{\kappa} + \omega$ |
| lower bound[2] | $\left(1 + \frac{\omega}{\sqrt{n}}\right)\sqrt{\kappa} + \omega$ | $\left(1 + \frac{\omega}{\sqrt{n}}\right)\sqrt{\kappa} + \omega$ | $(1 + \sqrt{\omega})\sqrt{\kappa} + \omega$ |
| LoCoDL | $\left(1 + \sqrt{\omega} + \frac{\omega}{\sqrt{n}}\right)\sqrt{\kappa} + \omega(1 + \frac{\omega}{n})$ | $(1 + \sqrt{\omega})\sqrt{\kappa} + \omega$ | $(1 + \sqrt{\omega})\sqrt{\kappa} + \omega$ |

[1]This is the complexity derived in the original paper Li et al. (2020b).
[2]This is the complexity derived by a refined analysis in the preprint He et al. (2023), where a matching lower bound is also derived.

Table 2: (Uplink) communication complexity in number of reals to reach $\epsilon$-accuracy for linearly-converging algorithms allowing for CC, with an optimal choice of unbiased compressors. We provide the leading asymptotic factor and ignore log factors such as $\log \epsilon^{-1}$. The state of the art is highlighted in green.

| Algorithm | complexity in # reals | case $n = \mathcal{O}(d)$ |
|---|---|---|
| DIANA | $(1 + \frac{d}{n})\kappa + d$ | $\frac{d}{n}\kappa + d$ |
| EF21 | $d\kappa$ | $d\kappa$ |
| 5GCS-CC | $\left(\sqrt{d} + \frac{d}{\sqrt{n}}\right)\sqrt{\kappa} + d$ | $\frac{d}{\sqrt{n}}\sqrt{\kappa} + d$ |
| ADIANA | $\left(1 + \frac{d}{\sqrt{n}}\right)\sqrt{\kappa} + d$ | $\frac{d}{\sqrt{n}}\sqrt{\kappa} + d$ |
| CompressedScaffnew | $\left(\sqrt{d} + \frac{d}{\sqrt{n}}\right)\sqrt{\kappa} + d$ | $\frac{d}{\sqrt{n}}\sqrt{\kappa} + d$ |
| FedCOMGATE | $d\kappa$ | $d\kappa$ |
| LoCoDL | $\left(\sqrt{d} + \frac{d}{\sqrt{n}}\right)\sqrt{\kappa} + d$ | $\frac{d}{\sqrt{n}}\sqrt{\kappa} + d$ |

another model estimate that converges to the solution $x^\star$ as well. We see the need of an additional model estimate that plays the role of an anchor for compression. This is the variable $y$ common to all clients in LoCoDL, which compress $x_i - y$ and send these compressed differences to the server.

**Combining the two ideas.** Accordingly, an equivalent reformulation of (1) is the consensus problem with $n + 1$ variables

$$\min_{x_1, \dots, x_n, y} \frac{1}{n} \sum_{i=1}^n f_i(x_i) + g(y) \text{ s.t. } x_1 = \cdots = x_n = y.$$

The primal–dual optimality conditions are $x_1 = \cdots = x_n = y$, $0 = \nabla f_i(x_i) - u_i \; \forall i \in [n]$, $0 = \nabla g(y) - v$, and $0 = u_1 + \cdots + u_n + nv$ (dual feasibility), for some dual variables $u_1, \dots, u_n, v$ introduced in LoCoDL, that always satisfy the dual feasibility condition.

## 2.2 Description of LoCoDL

LoCoDL is a randomized primal–dual algorithm, shown as Algorithm 1. At every iteration, for every $i \in [n]$ in parallel, Client $i$ first constructs a prediction $\hat{x}_i^t$ of its updated local model estimate, using a GD step with respect to $f_i$ corrected by the dual variable $u_i^t$. It also constructs a prediction $\hat{y}^t$ of the updated model estimate, using a GD step with respect to $g$ corrected by the dual variable $v^t$. Since $g$ is known by all clients, they all maintain and update identical copies of the variables $y$ and

---

**Algorithm 1** LoCoDL

---

1: **input:** stepsizes $\gamma > 0$, $\chi > 0$, $\rho > 0$; probability $p \in (0, 1]$; variance factor $\omega \geq 0$; local initial estimates $x_1^0, \ldots, x_n^0 \in \mathbb{R}^d$, initial estimate $y^0 \in \mathbb{R}^d$, initial control variates $u_1^0, \ldots, u_n^0 \in \mathbb{R}^d$ and $v \in \mathbb{R}^d$ such that $\frac{1}{n} \sum_{i=1}^{n} u_i^0 + v^0 = 0$.
2: **for** $t = 0, 1, \ldots$ **do**
3:   **for** $i = 1, \ldots, n$, at clients in parallel, **do**
4:     $\hat{x}_i^t := x_i^t - \gamma \nabla f_i(x_i^t) + \gamma u_i^t$
5:     $\hat{y}^t := y^t - \gamma \nabla g(y^t) + \gamma v^t$   // the clients store and update identical copies of $y^t, v^t, \hat{y}^t$
6:     flip a coin $\theta^t \in \{0, 1\}$ with $\mathrm{Prob}(\theta^t = 1) = p$
7:     **if** $\theta^t = 1$ **then**
8:       $d_i^t := \mathcal{C}_i^t\big(\hat{x}_i^t - \hat{y}^t\big)$
9:       send $d_i^t$ to the server
10:      at server: aggregate $\bar{d}^t := \frac{1}{2n} \sum_{j=1}^{n} d_j^t$ and broadcast $\bar{d}^t$ to all clients
11:      $x_i^{t+1} := (1 - \rho)\hat{x}_i^t + \rho(\hat{y}^t + \bar{d}^t)$
12:      $u_i^{t+1} := u_i^t + \frac{p\chi}{\gamma(1+2\omega)}\big(\bar{d}^t - d_i^t\big)$
13:      $y^{t+1} := \hat{y}^t + \rho\bar{d}^t$
14:      $v^{t+1} := v^t + \frac{p\chi}{\gamma(1+2\omega)}\bar{d}^t$
15:     **else**
16:       $x_i^{t+1} := \hat{x}_i^t, y^{t+1} = \hat{y}^t, u_i^{t+1} := u_i^t, v^{t+1} := v^t$
17:     **end if**
18:   **end for**
19: **end for**

---

$v$. If there is no communication, which is the case with probability $1 - p$, $x_i$ and $y$ are updated with these predicted estimates, and the dual variables $u_i$ and $v$ are unchanged. If communication occurs, which is the case with probability $p$, the clients compress the differences $\hat{x}_i^t - \hat{y}^t$ and send these compressed vectors to the server, which forms $\bar{d}^t$ equal to one half of their average. Then the variables $x_i$ are updated using a convex combination of the local predicted estimates $\hat{x}_i^t$ and the global but noisy estimate $\hat{y}^t + \bar{d}^t$. $y$ is updated similarly. Finally, the dual variables are updated using the compressed differences minus their weighted average, so that the dual feasibility condition remains satisfied. The model estimates $x_i^t, \hat{x}_i^t, y^t, \hat{y}^t$ all converge to $x^\star$, so that their differences, as well as the compressed differences as a consequence of (2), converge to zero. This is the key property that makes the algorithm variance-reduced. We consider the following assumption.

**Assumption 2.1** (class of compressors). In LoCoDL the compressors $\mathcal{C}_i^t$ are all in $\mathbb{U}(\omega)$ for some $\omega \geq 0$. Moreover, for every $i \in [n]$, $i' \in [n]$, $t \geq 0$, $t' \geq 0$, $\mathcal{C}_i^t$ and $\mathcal{C}_{i'}^{t'}$ are independent if $t \neq t'$ ($\mathcal{C}_i^t$ and $\mathcal{C}_{i'}^t$ at the same iteration $t$ need not be independent). We define $\omega_{\mathrm{av}} \geq 0$ such that for every $t \geq 0$, the collection $(\mathcal{C}_i^t)_{i=1}^n$ satisfies (3).

*Remark* 2.2 (partial participation). LoCoDL allows for a form of partial participation if we set $\rho = 1$. Indeed, in that case, at steps 11 and 13 of the algorithm, all local variables $x_i$ as well as the common variable $y$ are overwritten by the same up-to-date model $\hat{y}^t + \bar{d}^t$. So, it does not matter that for a non-participating client $i$ with $d_i^t = 0$, the $\hat{x}_i^{t'}$ were not computed for the $t' \leq t$ since its last participation, as they are not used in the process. However, a non-participating client should still update its local copy of $y$ at every iteration. This can be done when $\nabla g$ is much cheaper to compute that $\nabla f_i$, as is the case with $g = \frac{\mu}{2}\|\cdot\|^2$. A non-participating client can be completely idle for a certain period of time, but when it resumes participating, it should receive the last estimates of $x, y$ and $v$ from the server as it lost synchronization.

## 3   CONVERGENCE AND COMPLEXITY OF LoCoDL

**Theorem 3.1** (linear convergence of LoCoDL). *Suppose that Assumptions 1.1 and 2.1 hold. In* LoCoDL, *suppose that $0 < \gamma < \frac{2}{L}$, $2\rho - \rho^2(1 + \omega_{\mathrm{av}}) - \chi \geq 0$. For every $t \geq 0$, define the Lyapunov*

*function*

$$\Psi^t := \frac{1}{\gamma} \left( \sum_{i=1}^n \left\| x_i^t - x^\star \right\|^2 + n \left\| y^t - x^\star \right\|^2 \right) + \frac{\gamma(1 + 2\omega)}{p^2 \chi} \left( \sum_{i=1}^n \left\| u_i^t - u_i^\star \right\|^2 + n \left\| v^t - v^\star \right\|^2 \right),$$

(4)

*where $v^\star := \nabla g(x^\star)$ and $u_i^\star := \nabla f_i(x^\star)$. Then* LoCoDL *converges linearly: for every $t \geq 0$,*

$$\mathbb{E}\left[\Psi^t\right] \leq \tau^t \Psi^0, \quad \text{where} \quad \tau := \max\left( (1 - \gamma\mu)^2, (1 - \gamma L)^2, 1 - \frac{p^2 \chi}{1 + 2\omega} \right) < 1.$$

(5)

*In addition, for every $i \in [n]$, $(x_i^t)_{t \in \mathbb{N}}$ and $(y^t)_{t \in \mathbb{N}}$ converge to $x^\star$, $(u_i^t)_{t \in \mathbb{N}}$ converges to $u_i^\star$, and $(v^t)_{t \in \mathbb{N}}$ converges to $v^\star$, almost surely.*

We place ourselves in the conditions of Theorem 3.1. We observe that in (5), the larger $\chi$, the better, so given $\rho$ we should set $\chi = 2\rho - \rho^2(1 + \omega_{\text{av}})$. Then, choosing $\rho$ to maximize $\chi$ yields

$$\chi = \rho = \frac{1}{1 + \omega_{\text{av}}}.$$

(6)

We now study the complexity of LoCoDL with $\chi$ and $\rho$ chosen as in (6) and $\gamma = \Theta(\frac{1}{L})$. We remark that LoCoDL has the same rate $\tau^\sharp := \max(1 - \gamma\mu, \gamma L - 1)^2$ as mere distributed gradient descent, as long as $p^{-1}$, $\omega$ and $\omega_{\text{av}}$ are small enough to have $1 - \frac{p^2 \chi}{1+2\omega} \leq \tau^\sharp$. This is remarkable: communicating with a low frequency and compressed vectors does not harm convergence at all, until some threshold.

The iteration complexity of LoCoDL to reach $\epsilon$-accuracy, i.e. $\mathbb{E}[\Psi^t] \leq \epsilon\Psi^0$, is

$$\mathcal{O}\left( \left( \kappa + \frac{(1 + \omega_{\text{av}})(1 + \omega)}{p^2} \right) \log \epsilon^{-1} \right).$$

(7)

By choosing

$$p = \min\left( \sqrt{\frac{(1 + \omega_{\text{av}})(1 + \omega)}{\kappa}}, 1 \right),$$

(8)

the iteration complexity becomes $\mathcal{O}\left( \left(\kappa + \omega(1 + \omega_{\text{av}})\right) \log \epsilon^{-1} \right)$ and the communication complexity in number of communication rounds is $p$ times the iteration complexity, that is

$$\mathcal{O}\left( \left( \sqrt{\kappa(1 + \omega_{\text{av}})(1 + \omega)} + \omega(1 + \omega_{\text{av}}) \right) \log \epsilon^{-1} \right).$$

If the compressors are mutually independent, $\omega_{\text{av}} = \frac{\omega}{n}$ and the communication complexity can be equivalently written as

$$\mathcal{O}\left( \left( \left(1 + \sqrt{\omega} + \frac{\omega}{\sqrt{n}}\right) \sqrt{\kappa} + \omega\left(1 + \frac{\omega}{n}\right) \right) \log \epsilon^{-1} \right),$$

as shown in Table 1.

Let us consider the example of independent rand-$k$ compressors, for some $k \in [d]$. We have $\omega = \frac{d}{k} - 1$. Therefore, the communication complexity in numbers of reals is $k$ times the complexity in number of rounds; that is, $\mathcal{O}\left( \left( \left(\sqrt{kd} + \frac{d}{\sqrt{n}}\right) \sqrt{\kappa} + d\left(1 + \frac{d}{kn}\right) \right) \log \epsilon^{-1} \right)$. We can now choose $k$ to minimize this complexity: with $k = \lceil \frac{d}{n} \rceil$, it becomes $\mathcal{O}\left( \left( \left(\sqrt{d} + \frac{d}{\sqrt{n}}\right) \sqrt{\kappa} + d \right) \log \epsilon^{-1} \right)$, as shown in Table 2. Let us state this result:

**Corollary 3.2.** *In the conditions of Theorem 3.1, suppose in addition that the compressors $\mathcal{C}_i^t$ are independent* rand-$k$ *compressors with $k = \lceil \frac{d}{n} \rceil$. Suppose that $\gamma = \Theta(\frac{1}{L})$, $\chi = \rho = \frac{n}{n - 1 + d/k}$, and*

$$p = \min\left( \sqrt{\frac{dk(n - 1) + d^2}{nk^2\kappa}}, 1 \right).$$

(9)

*Then the uplink communication complexity in number of reals of* LoCoDL *is*

$$\mathcal{O}\left( \left( \sqrt{d}\sqrt{\kappa} + \frac{d\sqrt{\kappa}}{\sqrt{n}} + d \right) \log \epsilon^{-1} \right).$$

(10)

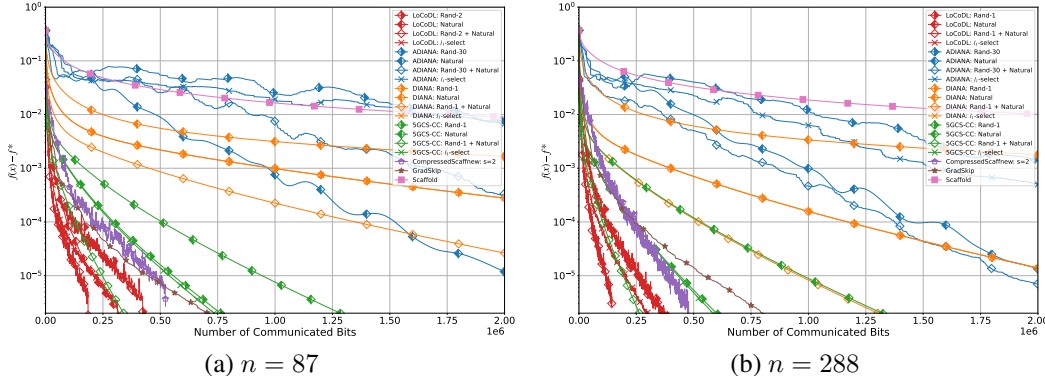

(a) $n = 87$ (b) $n = 288$

Figure 1: Comparison of several algorithms with several compressors on logistic regression with the 'a5a' dataset from the LibSVM, which has $d = 122$ and 6,414 data points. We chose different values of $n$ to illustrate the two regimes $n < d$ and $n > d$, as discussed at the end of Section 3.

This is the same complexity as CompressedScaffnew (Condat et al., 2022a). However, it is obtained with simple independent compressors, which is much more practical than the permutation-based compressors with shared randomness of CompressedScaffnew. Moreover, this complexity can be obtained with other types of compressors, and further reduced, when reasoning in number of bits and not only reals, by making use of quantization (Albasyoni et al., 2020), as we illustrate by experiments in the next section.

We can distinguish 2 regimes:

1. In the "large $d$ small $n$" regime, i.e. $n = \mathcal{O}(d)$, the communication complexity of LoCoDL in (10) becomes $\mathcal{O}\left(\left(\frac{d\sqrt{\kappa}}{\sqrt{n}} + d\right)\log \epsilon^{-1}\right)$. This is the state of the art, as reported in Table 2.

2. In the "large $n$ small $d$" regime, i.e. $n = \Omega(d)$, the communication complexity of LoCoDL in (10) becomes $\mathcal{O}\left(\left(\sqrt{d}\sqrt{\kappa} + d\right)\log \epsilon^{-1}\right)$. If $n$ is even larger with $n = \Omega(d^2)$, ADIANA achieves the even better complexity $\mathcal{O}\left((\sqrt{\kappa} + d)\log \epsilon^{-1}\right)$.

Yet, in the experiments we ran with different datasets and values of $d$, $n$, $\kappa$, LoCoDL outperforms the other algorithms, including ADIANA, in all cases.

### 3.1 THE CASE $g = 0$

We have assumed the presence of a function $g$ in Problem (1), whose gradient is called by all clients. In this section, we show that we can handle the case where such a function is not available. So, let us assume that we want to minimize $\frac{1}{n}\sum_{i=1}^{n} f_i$, with the functions $f_i$ satisfying Assumption 1.1. We now define the functions $\tilde{f}_i := f_i - \frac{\mu}{4}\left\|\cdot\right\|^2$ and $\tilde{g} := \frac{\mu}{4}\left\|\cdot\right\|^2$. They are all $\tilde{L}$-smooth and $\tilde{\mu}$-strongly convex, with $\tilde{L} := L - \frac{\mu}{2}$ and $\tilde{\mu} := \frac{\mu}{2}$. Moreover, it is equivalent to minimize $\frac{1}{n}\sum_{i=1}^{n} f_i$ or $\frac{1}{n}\sum_{i=1}^{n} \tilde{f}_i + \tilde{g}$. We can then apply LoCoDL to the latter problem. At Step 5, we simply have $y^t - \gamma\nabla\tilde{g}(y^t) = (1 - \frac{\gamma\mu}{2})y^t$. The rate in (5) applies with $L$ and $\mu$ replaced by $\tilde{L}$ and $\tilde{\mu}$, respectively. Since $\kappa \le \tilde{\kappa} := \frac{\tilde{L}}{\tilde{\mu}} \le 2\kappa$, the asymptotic complexities derived above also apply to this setting. Thus, the presence of $g$ in Problem (1) is not restrictive at all, as the only property of $g$ that matters is that it has the same amount of strong convexity as the $f_i$s.

## 4 EXPERIMENTS

We evaluate the performance of our proposed method LoCoDL and compare it with several other methods that also allow for CC and converge linearly to $x^\star$. We also include GradSkip (Maranjyan et al., 2022) and Scaffold (McMahan et al., 2017) in our comparisons. We focus on a regularized

logistic regression problem, which has the form (1) with

$$f_i(x) = \frac{1}{m} \sum_{s=1}^{m} \log\Big(1 + \exp\big(-b_{i,s} a_{i,s}^\top x\big)\Big) + \frac{\mu}{2} \|x\|^2 \tag{11}$$

and $g = \frac{\mu}{2}\|x\|^2$, where $n$ is the number of clients, $m$ is the number of data points per client, $a_{i,s} \in \mathbb{R}^d$ and $b_{i,s} \in \{-1, +1\}$ are the data samples, and $\mu$ is the regularization parameter, set so that $\kappa = 10^4$. For all algorithms other than LoCoDL, for which there is no function $g$, the functions $f_i$ in (11) have a twice higher $\mu$, so that the problem remains the same.

We considered several datasets from the LibSVM library (Chang & Lin, 2011) (3-clause BSD license). We show the results with the 'a5a' , 'diabetes', 'w1a' datasets in Figures 1, 2, 3, respectively. Other datasets are shown in the Appendix. We prepared each dataset by first shuffling it, then distributing it equally among the $n$ clients (since $m$ in (11) is an integer, the remaining datapoints were discarded). We used four different compression operators in the class $\mathbb{U}(\omega)$, for some $\omega \geq 0$:

• $\mathtt{rand}$-$k$ for some $k \in [d]$, which communicates $32k + k\lceil\log_2(d)\rceil$ bits. Indeed, the $k$ randomly chosen values are sent in the standard 32-bits IEEE floating-point format, and their locations are encoded with $k\lceil\log_2(d)\rceil$ additional bits. We have $\omega = \frac{d}{k} - 1$.

• Natural Compression (Horváth et al., 2022), a form of quantization in which floats are encoded into 9 bits instead of 32 bits. We have $\omega = \frac{1}{8}$.

• A combination of $\mathtt{rand}$-$k$ and Natural Compression, in which the $k$ chosen values are encoded into 9 bits, which yields a total of $9k + k\lceil\log_2(d)\rceil$ bits. We have $\omega = \frac{9d}{8k} - 1$.

• The $l_1$-selection compressor, defined as $C(x) = \mathrm{sign}(x_j)\|x\|_1 e_j$, where $j$ is chosen randomly in $[d]$, with the probability of choosing $j' \in [d]$ equal to $|x_{j'}|/\|x\|_1$, and $e_j$ is the $j$-th standard unit basis vector in $\mathbb{R}^d$. $\mathrm{sign}(x_j)\|x\|_1$ is sent as a 32-bits float and the location of $j$ is indicated with $\lceil\log_2(d)\rceil$, so that this compressor communicates $32 + \lceil\log_2(d)\rceil$ bits. Like with $\mathtt{rand}$-1, we have $\omega = d - 1$.

The compressors at different clients are independent, so that $\omega_{\mathrm{av}} = \frac{\omega}{n}$ in (3).

We can see that LoCoDL, when combined with $\mathtt{rand}$-$k$ and Natural Compression, converges faster than all other algorithms, with respect to the total number of communicated bits per client. We chose two different numbers $n$ of clients, one with $n < d$ and another one with $n > 2d$, since the compressor of CompressedScaffnew is different in the two cases $n < 2d$ and $n > 2d$ (Condat et al., 2022a). LoCoDL outperforms CompressedScaffnew in both cases. As expected, all methods exhibit faster convergence with larger $n$. Remarkably, ADIANA, which has the best theoretical complexity for large $n$, improves upon DIANA but is not competitive with the LT-based methods CompressedScaffnew, 5GCS-CC, and LoCoDL. This illustrates the power of doubly-accelerated methods based on a successful combination of LT and CC. In this class, our new proposed LoCoDL algorithm shines. For all algorithms, we used the theoretical parameter values given in their available convergence results (Corollary 3.2 for LoCoDL). We tried to tune the parameter values, such as $k$ in $\mathtt{rand}$-$k$ and the (average) number of local steps per round, but this only gave minor improvements. For instance, ADIANA in Figure 1 was a bit faster with the best value of $k = 20$ than with $k = 30$. Increasing the learning rate $\gamma$ led to inconsistent results, with sometimes divergence.

## 5 CONCLUSION

We have proposed LoCoDL, which combines a probabilistic Local Training mechanism similar to the one of Scaffnew and Communication Compression with a large class of unbiased compressors. This successful combination makes LoCoDL highly communication-efficient, with a doubly accelerated complexity with respect to the model dimension $d$ and the condition number of the functions. In practice, LoCoDL outperforms other algorithms, including ADIANA, which has an even better complexity in theory obtained from Nesterov acceleration and not Local Training. This again shows the relevance of the popular mechanism of Local Training, which has been widely adopted in Federated Learning. A venue for future work is to implement bidirectional compression (Liu et al., 2020; Philippenko & Dieuleveut, 2021; Dorfman et al., 2023). We will also investigate extensions of our method with calls to stochastic gradient estimates, with or without variance reduction, as well as partial participation. These two features have been proposed for Scaffnew in Malinovsky et al. (2022) and Condat et al. (2023), but they are challenging to combine with generic compression.

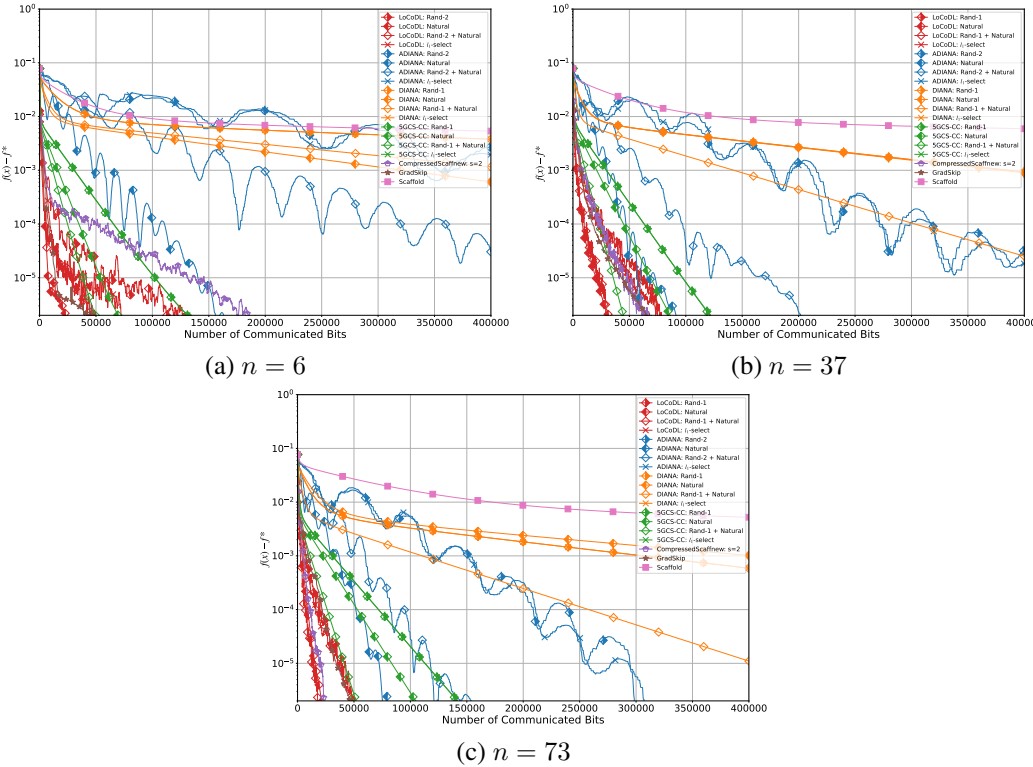

Figure 2: Comparison of several algorithms with several compressors on logistic regression with the 'diabetes' dataset from the LibSVM, which has $d = 8$ and 768 data points. We chose different values of $n$ to illustrate the three regimes $n < d$, $n > d$, $n > d^2$, as discussed at the end of Section 3.

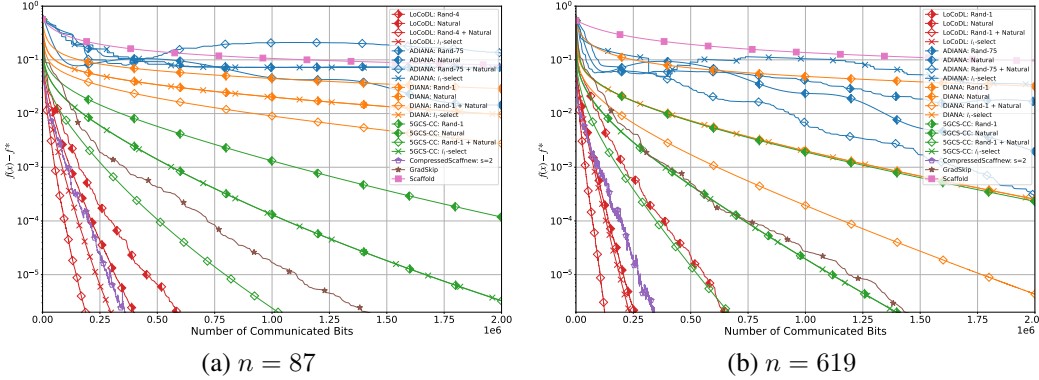

Figure 3: Comparison of several algorithms with various compressors on logistic regression with the 'w1a' dataset from the LibSVM, which has $d = 300$ and 2,477 data points. We chose different values of $n$ to illustrate the two regimes, $n < d$ and $n > d$, as discussed at the end of Section 3.

We have studied the strongly convex setting, because analyzing how the linear convergence rate depends on $d$, $\kappa$, $n$ provides valuable insights on the algorithmic mechanisms. It should be possible to derive sublinear convergence results in the general convex case, by studying the objective gap instead of the squared distance to the solution, as was done for CompressedScaffnew (Condat et al., 2022a) and TAMUNA (Condat et al., 2023). An analysis with nonconvex functions, however, would certainly require different proof techniques. In nonconvex settings, compression works well (Huang et al., 2022; Chen et al., 2024), but the properties of local training with variance reduction are less clear (Yi et al., 2024; Meinhardt et al., 2024).

ACKNOWLEDGMENTS

This work was supported by funding from King Abdullah University of Science and Technology (KAUST): i) KAUST Baseline Research Scheme, ii) Center of Excellence for Generative AI, under award number 5940, iii) SDAIA-KAUST Center of Excellence in Data Science and Artificial Intelligence.

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

# Appendix

## Contents

## A    Proof of Theorem 3.1

We define the Euclidean space $\mathcal{X} := \mathbb{R}^d$ and the product space $\boldsymbol{\mathcal{X}} := \mathcal{X}^{n+1}$ endowed with the weighted inner product

$$\langle \mathbf{x}, \mathbf{x}' \rangle_{\boldsymbol{\mathcal{X}}} := \sum_{i=1}^{n} \langle x_i, x_i' \rangle + n \langle y, y' \rangle, \quad \forall \mathbf{x} = (x_1, \ldots, x_n, y), \mathbf{x}' = (x_1', \ldots, x_n', y'). \quad (12)$$

We define the copy operator $\mathbf{1} : x \in \mathcal{X} \mapsto (x, \ldots, x, x) \in \boldsymbol{\mathcal{X}}$ and the linear operator

$$S : \mathbf{x} \in \boldsymbol{\mathcal{X}} \mapsto \mathbf{1}\bar{x}, \ \text{ with } \bar{x} = \frac{1}{2n}\left(\sum_{i=1}^{n} x_i + ny\right). \quad (13)$$

$S$ is the orthogonal projector in $\boldsymbol{\mathcal{X}}$ onto the consensus line $\{\mathbf{x} \in \boldsymbol{\mathcal{X}} \ : \ x_1 = \cdots = x_n = y\}$. We also define the linear operator

$$W := \mathrm{Id} - S : \mathbf{x} = (x_1, \ldots, x_n, y) \in \boldsymbol{\mathcal{X}} \mapsto (x_1 - \bar{x}, \ldots, x_n - \bar{x}, y - \bar{x}), \ \text{with } \bar{x} = \frac{1}{2n}\left(\sum_{i=1}^{n} x_i + ny\right),$$
$$(14)$$

where Id denotes the identity. $W$ is the orthogonal projector in $\boldsymbol{\mathcal{X}}$ onto the hyperplane $\{\mathbf{x} \in \boldsymbol{\mathcal{X}} \ : \ x_1 + \cdots + x_n + ny = 0\}$, which is orthogonal to the consensus line. As such, it is self-adjoint, positive semidefinite, its eigenvalues are $(1, \ldots, 1, 0)$, its kernel is the consensus line, and its spectral

norm is 1. Also, $W^2 = W$. Note that we can write $W$ in terms of the differences $d_i = x_i - y$ and $\bar{d} = \frac{1}{2n} \sum_{i=1}^{n} d_i$:

$$W : \mathbf{x} = (x_1, \ldots, x_n, y) \mapsto (d_1 - \bar{d}, \ldots, d_n - \bar{d}, -\bar{d}). \tag{15}$$

Since for every $\mathbf{x} = (x_1, \ldots, x_n, y)$, $W\mathbf{x} = \mathbf{0} := (0, \ldots, 0, 0)$ if and only if $x_1 = \cdots = x_n = y$, we can reformulate the problem (1) as

$$\min_{\mathbf{x} = (x_1, \ldots, x_n, y) \in \boldsymbol{\mathcal{X}}} \mathbf{f}(\mathbf{x}) \quad \text{s.t.} \quad W\mathbf{x} = \mathbf{0}, \tag{16}$$

where $\mathbf{f}(\mathbf{x}) := \sum_{i=1}^{n} f_i(x_i) + ng(y)$. Note that in $\boldsymbol{\mathcal{X}}$, $\mathbf{f}$ is $L$-smooth and $\mu$-strongly convex, and $\nabla \mathbf{f}(\mathbf{x}) = (\nabla f_1(x_1), \ldots \nabla f_n(x_n), \nabla g(y))$.

Let $t \geq 0$. We also introduce vector notations for the variables of the algorithm: $\mathbf{x}^t := (x_1^t, \ldots, x_n^t, y^t)$, $\hat{\mathbf{x}}^t := (\hat{x}_1^t, \ldots, \hat{x}_n^t, \hat{y}^t)$, $\mathbf{u}^t := (u_1^t, \ldots, u_n^t, v^t)$, $\mathbf{u}^\star := (u_1^\star, \ldots, u_n^\star, v^\star)$, $\mathbf{w}^t := \mathbf{x}^t - \gamma \nabla \mathbf{f}(\mathbf{x}^t)$, $\mathbf{w}^\star := \mathbf{x}^\star - \gamma \nabla \mathbf{f}(\mathbf{x}^\star)$, where $\mathbf{x}^\star := \mathbf{1}x^\star$ is the unique solution to (16). We also define $\bar{x}^t := \frac{1}{2n}(\sum_{i=1}^{n} \hat{x}_i^t + n\hat{y}^t)$ and $\lambda := \frac{p\chi}{\gamma(1+2\omega)}$.

Then we can write the iteration of LoCoDL as

$$
\left|
\begin{array}{l}
\hat{\mathbf{x}}^t := \mathbf{x}^t - \gamma \nabla \mathbf{f}(\mathbf{x}^t) + \gamma \mathbf{u}^t = \mathbf{w}^t + \gamma \mathbf{u}^t \\
\text{flip a coin } \theta^t \in \{0, 1\} \text{ with } \mathrm{Prob}(\theta^t = 1) = p \\
\textbf{if } \theta^t = 1 \\
\quad \mathbf{d}^t := (\mathcal{C}_1^t(\hat{x}_1^t - \hat{y}^t), \ldots, \mathcal{C}_n^t(\hat{x}_n^t - \hat{y}^t), 0) \\
\quad \bar{d}^t := \frac{1}{2n} \sum_{j=1}^{n} d_j^t \\
\quad \mathbf{x}^{t+1} := (1 - \rho)\hat{\mathbf{x}}^t + \rho\mathbf{1}(\hat{y}^t + \bar{d}^t) \\
\quad \mathbf{u}^{t+1} := \mathbf{u}^t + \lambda(\mathbf{1}\bar{d}^t - \mathbf{d}^t) = \mathbf{u}^t - \lambda W\mathbf{d}^t \\
\textbf{else} \\
\quad \mathbf{x}^{t+1} := \hat{\mathbf{x}}^t \\
\quad \mathbf{u}^{t+1} := \mathbf{u}^t \\
\textbf{end if}
\end{array}
\right. \tag{17}
$$

We denote by $\mathcal{F}^t$ the $\sigma$-algebra generated by the collection of $\boldsymbol{\mathcal{X}}$-valued random variables $\mathbf{x}^0, \mathbf{u}^0, \ldots, \mathbf{x}^t, \mathbf{u}^t$.

Since we suppose that $S\mathbf{u}^0 = \mathbf{0}$ and we have $SW\mathbf{d}^{t'} = \mathbf{0}$ in the update of $\mathbf{u}$, we have $S\mathbf{u}^{t'} = \mathbf{0}$ for every $t' \geq 0$.

If $\theta^t = 1$, we have

$$
\begin{aligned}
\left\| \mathbf{u}^{t+1} - \mathbf{u}^\star \right\|_{\boldsymbol{\mathcal{X}}}^2 &= \left\| \mathbf{u}^t - \mathbf{u}^\star \right\|_{\boldsymbol{\mathcal{X}}}^2 + \lambda^2 \left\| W\mathbf{d}^t \right\|_{\boldsymbol{\mathcal{X}}}^2 - 2\lambda \langle \mathbf{u}^t - \mathbf{u}^\star, W\mathbf{d}^t \rangle_{\boldsymbol{\mathcal{X}}} \\
&= \left\| \mathbf{u}^t - \mathbf{u}^\star \right\|_{\boldsymbol{\mathcal{X}}}^2 + \lambda^2 \left\| \mathbf{d}^t \right\|_{\boldsymbol{\mathcal{X}}}^2 - \lambda^2 \left\| S\mathbf{d}^t \right\|_{\boldsymbol{\mathcal{X}}}^2 - 2\lambda \langle \mathbf{u}^t - \mathbf{u}^\star, \mathbf{d}^t \rangle_{\boldsymbol{\mathcal{X}}},
\end{aligned}
$$

because $S\mathbf{u}^t = S\mathbf{u}^\star = \mathbf{0}$, so that $\langle \mathbf{u}^t - \mathbf{u}^\star, S\mathbf{d}^t \rangle_{\boldsymbol{\mathcal{X}}} = 0$.

The variance inequality (2) satisfied by the compressors $\mathcal{C}_i^t$ is equivalent to $\mathbb{E}\left[ \|\mathcal{C}_i^t(x)\|^2 \right] \leq (1 + \omega) \|x\|^2$, so that

$$\mathbb{E}\left[ \left\| \mathbf{d}^t \right\|_{\boldsymbol{\mathcal{X}}}^2 \mid \mathcal{F}^t, \theta^t = 1 \right] \leq (1 + \omega) \left\| \hat{\mathbf{x}}^t - \mathbf{1}\hat{y}^t \right\|_{\boldsymbol{\mathcal{X}}}^2.$$

Also,

$$\mathbb{E}\left[ \mathbf{d}^t \mid \mathcal{F}^t, \theta^t = 1 \right] = \hat{\mathbf{x}}^t - \mathbf{1}\hat{y}^t.$$

Thus,

$$
\begin{aligned}
\mathbb{E}\left[ \left\| \mathbf{u}^{t+1} - \mathbf{u}^\star \right\|_{\boldsymbol{\mathcal{X}}}^2 \mid \mathcal{F}^t \right] &= (1 - p) \left\| \mathbf{u}^t - \mathbf{u}^\star \right\|_{\boldsymbol{\mathcal{X}}}^2 + p\mathbb{E}\left[ \left\| \mathbf{u}^{t+1} - \mathbf{u}^\star \right\|_{\boldsymbol{\mathcal{X}}}^2 \mid \mathcal{F}^t, \theta^t = 1 \right] \\
&\leq \left\| \mathbf{u}^t - \mathbf{u}^\star \right\|_{\boldsymbol{\mathcal{X}}}^2 + p\lambda^2(1 + \omega) \left\| \hat{\mathbf{x}}^t - \mathbf{1}\hat{y}^t \right\|_{\boldsymbol{\mathcal{X}}}^2 - p\lambda^2\mathbb{E}\left[ \left\| S\mathbf{d}^t \right\|_{\boldsymbol{\mathcal{X}}}^2 \mid \mathcal{F}^t, \theta^t = 1 \right] \\
&\quad - 2p\lambda \langle \mathbf{u}^t - \mathbf{u}^\star, \hat{\mathbf{x}}^t - \mathbf{1}\hat{y}^t \rangle_{\boldsymbol{\mathcal{X}}} \\
&= \left\| \mathbf{u}^t - \mathbf{u}^\star \right\|_{\boldsymbol{\mathcal{X}}}^2 + p\lambda^2(1 + \omega) \left\| \hat{\mathbf{x}}^t - \mathbf{1}\hat{y}^t \right\|_{\boldsymbol{\mathcal{X}}}^2 - p\lambda^2\mathbb{E}\left[ \left\| S\mathbf{d}^t \right\|_{\boldsymbol{\mathcal{X}}}^2 \mid \mathcal{F}^t, \theta^t = 1 \right] \\
&\quad - 2p\lambda \langle \mathbf{u}^t - \mathbf{u}^\star, \hat{\mathbf{x}}^t \rangle_{\boldsymbol{\mathcal{X}}}.
\end{aligned}
$$

Moreover, $\mathbb{E}\left[\left\|S\mathbf{d}^t\right\|_{\boldsymbol{\mathcal{X}}}^2 \mid \mathcal{F}^t, \theta^t = 1\right] \geq \left\|\mathbb{E}[S\mathbf{d}^t \mid \mathcal{F}^t, \theta^t = 1]\right\|_{\boldsymbol{\mathcal{X}}}^2 = \left\|S\hat{\mathbf{x}}^t - \mathbf{1}\hat{y}^t\right\|_{\boldsymbol{\mathcal{X}}}^2$ and $\left\|\hat{\mathbf{x}}^t - \mathbf{1}\hat{y}^t\right\|_{\boldsymbol{\mathcal{X}}}^2 = \left\|S\hat{\mathbf{x}}^t - \mathbf{1}\hat{y}^t\right\|_{\boldsymbol{\mathcal{X}}}^2 + \left\|W\hat{\mathbf{x}}^t\right\|_{\boldsymbol{\mathcal{X}}}^2$, so that

$$
\begin{aligned}
\mathbb{E}\left[\left\|\mathbf{u}^{t+1} - \mathbf{u}^\star\right\|_{\boldsymbol{\mathcal{X}}}^2 \mid \mathcal{F}^t\right] &\leq \left\|\mathbf{u}^t - \mathbf{u}^\star\right\|_{\boldsymbol{\mathcal{X}}}^2 + p\lambda^2(1+\omega)\left\|\hat{\mathbf{x}}^t - \mathbf{1}\hat{y}^t\right\|_{\boldsymbol{\mathcal{X}}}^2 - p\lambda^2 \left\|S\hat{\mathbf{x}}^t - \mathbf{1}\hat{y}^t\right\|^2 \\
&\quad - 2p\lambda\langle \mathbf{u}^t - \mathbf{u}^\star, \hat{\mathbf{x}}^t\rangle_{\boldsymbol{\mathcal{X}}} \\
&= \left\|\mathbf{u}^t - \mathbf{u}^\star\right\|_{\boldsymbol{\mathcal{X}}}^2 + p\lambda^2\omega\left\|\hat{\mathbf{x}}^t - \mathbf{1}\hat{y}^t\right\|_{\boldsymbol{\mathcal{X}}}^2 + p\lambda^2 \left\|W\hat{\mathbf{x}}^t\right\|^2 - 2p\lambda\langle \mathbf{u}^t - \mathbf{u}^\star, \hat{\mathbf{x}}^t\rangle_{\boldsymbol{\mathcal{X}}}.
\end{aligned}
$$

From the Peter–Paul inequality $\|a + b\|^2 \leq 2\|a\|^2 + 2\|b\|^2$ for any $a$ and $b$, we have

$$
\begin{aligned}
\left\|\hat{\mathbf{x}}^t - \mathbf{1}\hat{y}^t\right\|_{\boldsymbol{\mathcal{X}}}^2 &= \sum_{i=1}^n \left\|\hat{x}_i^t - \hat{y}^t\right\|^2 = \sum_{i=1}^n \left\|(\hat{x}_i^t - \bar{x}^t) - (\hat{y}^t - \bar{x}^t)\right\|^2 \\
&\leq \sum_{i=1}^n \left(2\left\|\hat{x}_i^t - \bar{x}^t)\right\|^2 + 2\left\|\hat{y}^t - \bar{x}^t\right\|^2\right) \\
&= 2\left(\sum_{i=1}^n \left\|\hat{x}_i^t - \bar{x}^t)\right\|^2 + n\left\|\hat{y}^t - \bar{x}^t\right\|^2\right) \\
&= 2\left\|\hat{\mathbf{x}}^t - \mathbf{1}\bar{x}^t\right\|_{\boldsymbol{\mathcal{X}}}^2 = 2\left\|W\hat{\mathbf{x}}^t\right\|_{\boldsymbol{\mathcal{X}}}^2.
\end{aligned} \tag{18}
$$

Hence,

$$
\mathbb{E}\left[\left\|\mathbf{u}^{t+1} - \mathbf{u}^\star\right\|_{\boldsymbol{\mathcal{X}}}^2 \mid \mathcal{F}^t\right] \leq \left\|\mathbf{u}^t - \mathbf{u}^\star\right\|_{\boldsymbol{\mathcal{X}}}^2 + p\lambda^2(1+2\omega)\left\|W\hat{\mathbf{x}}^t\right\|_{\boldsymbol{\mathcal{X}}}^2 - 2p\lambda\langle \mathbf{u}^t - \mathbf{u}^\star, \hat{\mathbf{x}}^t\rangle_{\boldsymbol{\mathcal{X}}}.
$$

On the other hand,

$$
\begin{aligned}
\mathbb{E}\left[\left\|\mathbf{x}^{t+1} - \mathbf{x}^\star\right\|_{\boldsymbol{\mathcal{X}}}^2 \mid \mathcal{F}^t, \theta = 1\right] &= (1-\rho)^2 \left\|\hat{\mathbf{x}}^t - \mathbf{x}^\star\right\|_{\boldsymbol{\mathcal{X}}}^2 + \rho^2\mathbb{E}\left[\left\|\mathbf{1}(\hat{y}^t + \bar{d}^t) - \mathbf{x}^\star\right\|_{\boldsymbol{\mathcal{X}}}^2 \mid \mathcal{F}^t, \theta = 1\right] \\
&\quad + 2\rho(1-\rho)\left\langle\hat{\mathbf{x}}^t - \mathbf{x}^\star, \mathbf{1}\left(\hat{y}^t + \mathbb{E}\left[\bar{d}^t \mid \mathcal{F}^t, \theta = 1\right]\right) - \mathbf{x}^\star\right\rangle_{\boldsymbol{\mathcal{X}}}.
\end{aligned}
$$

We have $\mathbb{E}\left[\bar{d}^t \mid \mathcal{F}^t, \theta = 1\right] = \frac{1}{2n}\sum_{i=1}^n \hat{x}_i^t - \frac{1}{2}\hat{y}^t = \bar{x}^t - \hat{y}^t$, so that

$$
\mathbf{1}\left(\hat{y}^t + \mathbb{E}\left[\bar{d}^t \mid \mathcal{F}^t, \theta = 1\right]\right) = \mathbf{1}\bar{x}^t = S\hat{\mathbf{x}}^t.
$$

In addition,

$$
\left\langle\hat{\mathbf{x}}^t - \mathbf{x}^\star, S\hat{\mathbf{x}}^t - \mathbf{x}^\star\right\rangle_{\boldsymbol{\mathcal{X}}} = \left\langle\hat{\mathbf{x}}^t - \mathbf{x}^\star, S(\hat{\mathbf{x}}^t - \mathbf{x}^\star)\right\rangle_{\boldsymbol{\mathcal{X}}} = \left\|S(\hat{\mathbf{x}}^t - \mathbf{x}^\star)\right\|_{\boldsymbol{\mathcal{X}}}^2.
$$

Moreover,

$$
\begin{aligned}
\mathbb{E}\left[\left\|\mathbf{1}(\hat{y}^t + \bar{d}^t) - \mathbf{x}^\star\right\|_{\boldsymbol{\mathcal{X}}}^2 \mid \mathcal{F}^t, \theta = 1\right] &= \left\|\mathbf{1}\left(\hat{y}^t + \mathbb{E}\left[\bar{d}^t \mid \mathcal{F}^t, \theta = 1\right]\right) - \mathbf{x}^\star\right\|_{\boldsymbol{\mathcal{X}}}^2 \\
&\quad + \mathbb{E}\left[\left\|\mathbf{1}\left(\bar{d}^t - \mathbb{E}\left[\bar{d}^t \mid \mathcal{F}^t, \theta = 1\right]\right)\right\|_{\boldsymbol{\mathcal{X}}}^2 \mid \mathcal{F}^t, \theta = 1\right] \\
&= \left\|S\hat{\mathbf{x}}^t - \mathbf{x}^\star\right\|_{\boldsymbol{\mathcal{X}}}^2 \\
&\quad + 2n\mathbb{E}\left[\left\|\bar{d}^t - \mathbb{E}\left[\bar{d}^t \mid \mathcal{F}^t, \theta = 1\right]\right\|^2 \mid \mathcal{F}^t, \theta = 1\right]
\end{aligned}
$$

and, using (3),

$$
\begin{aligned}
\mathbb{E}\left[\left\|\bar{d}^t - \mathbb{E}\left[\bar{d}^t \mid \mathcal{F}^t, \theta = 1\right]\right\|^2 \mid \mathcal{F}^t, \theta = 1\right] &\leq \frac{\omega_{\mathrm{av}}}{4n}\sum_{i=1}^n \left\|\hat{x}_i^t - \hat{y}^t\right\|^2 \\
&\leq \frac{\omega_{\mathrm{av}}}{2n}\left\|W\hat{\mathbf{x}}^t\right\|_{\boldsymbol{\mathcal{X}}}^2,
\end{aligned}
$$

where the second inequality follows from (18). Hence,

$$\mathbb{E}\Big[\big\|\mathbf{x}^{t+1} - \mathbf{x}^\star\big\|_{\boldsymbol{\mathcal{X}}}^2 \mid \mathcal{F}^t, \theta = 1\Big] \le (1-\rho)^2 \big\|\hat{\mathbf{x}}^t - \mathbf{x}^\star\big\|_{\boldsymbol{\mathcal{X}}}^2 + \rho^2 \big\|S\hat{\mathbf{x}}^t - \mathbf{x}^\star\big\|_{\boldsymbol{\mathcal{X}}}^2 + \rho^2 \omega_{\mathrm{av}} \big\|W\hat{\mathbf{x}}^t\big\|_{\boldsymbol{\mathcal{X}}}^2$$

$$+ 2\rho(1-\rho) \big\|S(\hat{\mathbf{x}}^t - \mathbf{x}^\star)\big\|_{\boldsymbol{\mathcal{X}}}^2$$

$$= (1-\rho)^2 \big\|\hat{\mathbf{x}}^t - \mathbf{x}^\star\big\|_{\boldsymbol{\mathcal{X}}}^2 + \rho^2 \omega_{\mathrm{av}} \big\|W\hat{\mathbf{x}}^t\big\|_{\boldsymbol{\mathcal{X}}}^2$$

$$+ (2\rho - \rho^2) \big\|S(\hat{\mathbf{x}}^t - \mathbf{x}^\star)\big\|_{\boldsymbol{\mathcal{X}}}^2$$

$$= (1-\rho)^2 \big\|\hat{\mathbf{x}}^t - \mathbf{x}^\star\big\|_{\boldsymbol{\mathcal{X}}}^2 + \rho^2 \omega_{\mathrm{av}} \big\|W\hat{\mathbf{x}}^t\big\|_{\boldsymbol{\mathcal{X}}}^2$$

$$+ (2\rho - \rho^2) \Big(\big\|\hat{\mathbf{x}}^t - \mathbf{x}^\star\big\|_{\boldsymbol{\mathcal{X}}}^2 - \big\|W\hat{\mathbf{x}}^t\big\|_{\boldsymbol{\mathcal{X}}}^2\Big)$$

$$= \big\|\hat{\mathbf{x}}^t - \mathbf{x}^\star\big\|_{\boldsymbol{\mathcal{X}}}^2 - \big(2\rho - \rho^2 - \rho^2 \omega_{\mathrm{av}}\big) \big\|W\hat{\mathbf{x}}^t\big\|_{\boldsymbol{\mathcal{X}}}^2$$

and

$$\mathbb{E}\Big[\big\|\mathbf{x}^{t+1} - \mathbf{x}^\star\big\|_{\boldsymbol{\mathcal{X}}}^2 \mid \mathcal{F}^t\Big] = (1-p) \big\|\hat{\mathbf{x}}^t - \mathbf{x}^\star\big\|_{\boldsymbol{\mathcal{X}}}^2 + p\mathbb{E}\Big[\big\|\mathbf{x}^{t+1} - \mathbf{x}^\star\big\|_{\boldsymbol{\mathcal{X}}}^2 \mid \mathcal{F}^t, \theta^t = 1\Big]$$

$$\le \big\|\hat{\mathbf{x}}^t - \mathbf{x}^\star\big\|_{\boldsymbol{\mathcal{X}}}^2 - p\big(2\rho - \rho^2(1 + \omega_{\mathrm{av}})\big) \big\|W\hat{\mathbf{x}}^t\big\|_{\boldsymbol{\mathcal{X}}}^2.$$

Furthermore,

$$\big\|\hat{\mathbf{x}}^t - \mathbf{x}^\star\big\|_{\boldsymbol{\mathcal{X}}}^2 = \big\|\mathbf{w}^t - \mathbf{w}^\star\big\|_{\boldsymbol{\mathcal{X}}}^2 + \gamma^2 \big\|\mathbf{u}^t - \mathbf{u}^\star\big\|_{\boldsymbol{\mathcal{X}}}^2 + 2\gamma\langle\mathbf{w}^t - \mathbf{w}^\star, \mathbf{u}^t - \mathbf{u}^\star\rangle_{\boldsymbol{\mathcal{X}}}$$

$$= \big\|\mathbf{w}^t - \mathbf{w}^\star\big\|_{\boldsymbol{\mathcal{X}}}^2 - \gamma^2 \big\|\mathbf{u}^t - \mathbf{u}^\star\big\|_{\boldsymbol{\mathcal{X}}}^2 + 2\gamma\langle\hat{\mathbf{x}}^t - \mathbf{x}^\star, \mathbf{u}^t - \mathbf{u}^\star\rangle_{\boldsymbol{\mathcal{X}}}$$

$$= \big\|\mathbf{w}^t - \mathbf{w}^\star\big\|_{\boldsymbol{\mathcal{X}}}^2 - \gamma^2 \big\|\mathbf{u}^t - \mathbf{u}^\star\big\|_{\boldsymbol{\mathcal{X}}}^2 + 2\gamma\langle\hat{\mathbf{x}}^t, \mathbf{u}^t - \mathbf{u}^\star\rangle_{\boldsymbol{\mathcal{X}}},$$

which yields

$$\mathbb{E}\Big[\big\|\mathbf{x}^{t+1} - \mathbf{x}^\star\big\|_{\boldsymbol{\mathcal{X}}}^2 \mid \mathcal{F}^t\Big] \le \big\|\mathbf{w}^t - \mathbf{w}^\star\big\|_{\boldsymbol{\mathcal{X}}}^2 - \gamma^2 \big\|\mathbf{u}^t - \mathbf{u}^\star\big\|_{\boldsymbol{\mathcal{X}}}^2 + 2\gamma\langle\hat{\mathbf{x}}^t, \mathbf{u}^t - \mathbf{u}^\star\rangle_{\boldsymbol{\mathcal{X}}}$$

$$- p\big(2\rho - \rho^2(1 + \omega_{\mathrm{av}})\big) \big\|W\hat{\mathbf{x}}^t\big\|_{\boldsymbol{\mathcal{X}}}^2.$$

Hence, with $\lambda = \frac{p\chi}{\gamma(1+2\omega)}$,

$$\frac{1}{\gamma}\mathbb{E}\Big[\big\|\mathbf{x}^{t+1} - \mathbf{x}^\star\big\|_{\boldsymbol{\mathcal{X}}}^2 \mid \mathcal{F}^t\Big] + \frac{\gamma(1+2\omega)}{p^2\chi}\mathbb{E}\Big[\big\|\mathbf{u}^{t+1} - \mathbf{u}^\star\big\|_{\boldsymbol{\mathcal{X}}}^2 \mid \mathcal{F}^t\Big]$$

$$\le \frac{1}{\gamma} \big\|\mathbf{w}^t - \mathbf{w}^\star\big\|_{\boldsymbol{\mathcal{X}}}^2 - \gamma \big\|\mathbf{u}^t - \mathbf{u}^\star\big\|_{\boldsymbol{\mathcal{X}}}^2 + 2\langle\hat{\mathbf{x}}^t, \mathbf{u}^t - \mathbf{u}^\star\rangle_{\boldsymbol{\mathcal{X}}} - \frac{p}{\gamma}\big(2\rho - \rho^2(1 + \omega_{\mathrm{av}})\big) \big\|W\hat{\mathbf{x}}^t\big\|_{\boldsymbol{\mathcal{X}}}^2$$

$$+ \frac{\gamma(1+2\omega)}{p^2\chi} \big\|\mathbf{u}^t - \mathbf{u}^\star\big\|_{\boldsymbol{\mathcal{X}}}^2 + \frac{p\chi}{\gamma} \big\|W\hat{\mathbf{x}}^t\big\|_{\boldsymbol{\mathcal{X}}}^2 - 2\langle\mathbf{u}^t - \mathbf{u}^\star, \hat{\mathbf{x}}^t\rangle_{\boldsymbol{\mathcal{X}}}$$

$$= \frac{1}{\gamma} \big\|\mathbf{w}^t - \mathbf{w}^\star\big\|_{\boldsymbol{\mathcal{X}}}^2 + \frac{\gamma(1+2\omega)}{p^2\chi} \left(1 - \frac{p^2\chi}{1+2\omega}\right) \big\|\mathbf{u}^t - \mathbf{u}^\star\big\|_{\boldsymbol{\mathcal{X}}}^2$$

$$- \frac{p}{\gamma}\big(2\rho - \rho^2(1 + \omega_{\mathrm{av}}) - \chi\big) \big\|W\hat{\mathbf{x}}^t\big\|_{\boldsymbol{\mathcal{X}}}^2.$$

Therefore, assuming that $2\rho - \rho^2(1 + \omega_{\mathrm{av}}) - \chi \ge 0$,

$$\mathbb{E}\big[\Psi^{t+1} \mid \mathcal{F}^t\big] \le \frac{1}{\gamma} \big\|\mathbf{w}^t - \mathbf{w}^\star\big\|_{\boldsymbol{\mathcal{X}}}^2 + \left(1 - \frac{p^2\chi}{1+2\omega}\right) \frac{\gamma(1+2\omega)}{p^2\chi} \big\|\mathbf{u}^t - \mathbf{u}^\star\big\|_{\boldsymbol{\mathcal{X}}}^2.$$

According to Condat & Richtárik (2023, Lemma 1),

$$\big\|\mathbf{w}^t - \mathbf{w}^\star\big\|_{\boldsymbol{\mathcal{X}}}^2 = \big\|(\mathrm{Id} - \gamma\nabla\mathbf{f})\mathbf{x}^t - (\mathrm{Id} - \gamma\nabla\mathbf{f})\mathbf{x}^\star\big\|_{\boldsymbol{\mathcal{X}}}^2$$

$$\le \max(1 - \gamma\mu, \gamma L - 1)^2 \big\|\mathbf{x}^t - \mathbf{x}^\star\big\|_{\boldsymbol{\mathcal{X}}}^2.$$

Hence,

$$\mathbb{E}\big[\Psi^{t+1} \mid \mathcal{F}^t\big] \le \max\left((1 - \gamma\mu)^2, (1 - \gamma L)^2, 1 - \frac{p^2\chi}{1+2\omega}\right) \Psi^t. \tag{19}$$

Using the tower rule, we can unroll the recursion in (19) to obtain the unconditional expectation of $\Psi^{t+1}$.

Using classical results on supermartingale convergence (Bertsekas, 2015, Proposition A.4.5), it follows from (19) that $\Psi^t \to 0$ almost surely. Almost sure convergence of $\mathbf{x}^t$ and $\mathbf{u}^t$ follows.

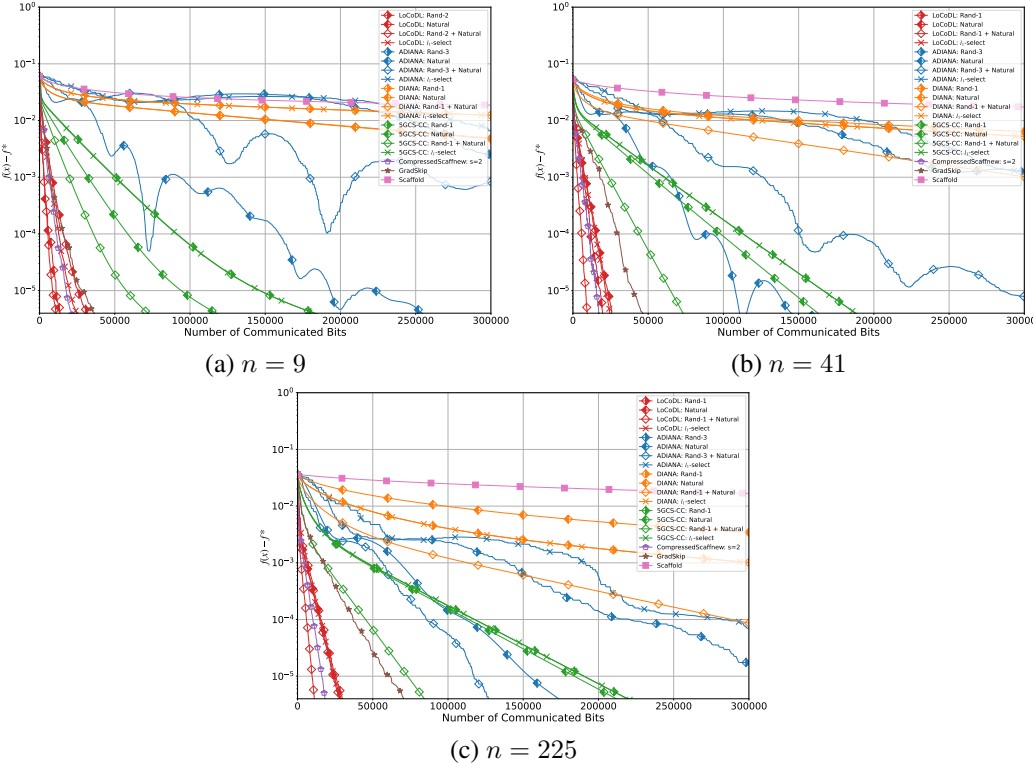

(a) $n = 9$

(b) $n = 41$

(c) $n = 225$

Figure 4: Comparison of several algorithms with various compressors on logistic regression with the 'australian' dataset from the LibSVM, which has $d = 14$ and 690 data points. We chose different values of $n$ to illustrate the three regimes: $n < d$, $n > d$, $n > d^2$, as discussed at the end of Section 3.

## B    ADDITIONAL EXPERIMENTS

The results with the 'australian' and 'covtype.binary' datasets from the LibSVM library, for the same logistic regression problem as in Section 4 with $\kappa = 10^4$, are shown in Figures 4 and 5. Finally, we also run experiments on MNIST dataset (LeCun et al., 1998) in Figure 6. LoCoDL consistently outperforms the other algorithms in terms of communication efficiency.

In an additional experiment, we investigate how heterogeneity of the functions influences the convergence. We consider logistic regression as above, but with synthetic data sampled from the Dirichlet distribution of parameter $\alpha$. If $\alpha$ is small, the Dirichlet distribution becomes similar to the uniform distribution over the simplex, which corresponds to the heterogeneous case where there is no similarity between the data. If $\alpha$ is large, the samples of the Dirichlet distribution tend to be similar to each other and concentrated around the middle point $(1/d, \ldots, 1/d)$ of the simplex. We set $n = 100$ and $d = 10$, and a single random sample is assigned to each client. The results are shown in Figure 7, for $\alpha = 1$ and $\alpha = 10$. With these values of $n$ and $d$, ADIANA has a better theoretical complexity than LoCoDL. However, in practice, we observe that LoCoDL again outperforms ADIANA. For both methods, joint sparsification and quantization with rand-1 and natural compression performs best. There is no significant difference depending on the value of $\alpha$.

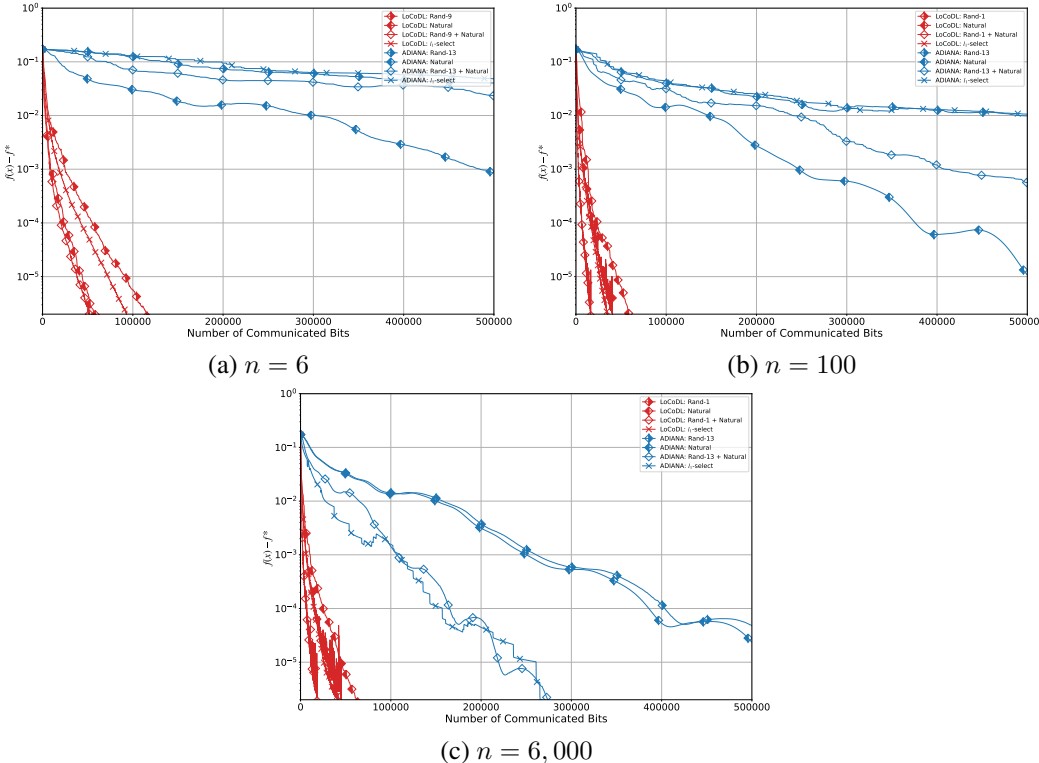

Figure 5: comparison of LoCoDL and ADIANA using several compressors for logistic regression with the 'covtype.binary' dataset from the LibSVM, which has $d = 54$ and 581,010 data points. We chose different values of $n$ to illustrate the three regimes $n < d$, $n > d$, $n > d^2$, as discussed at the end of Section 3.

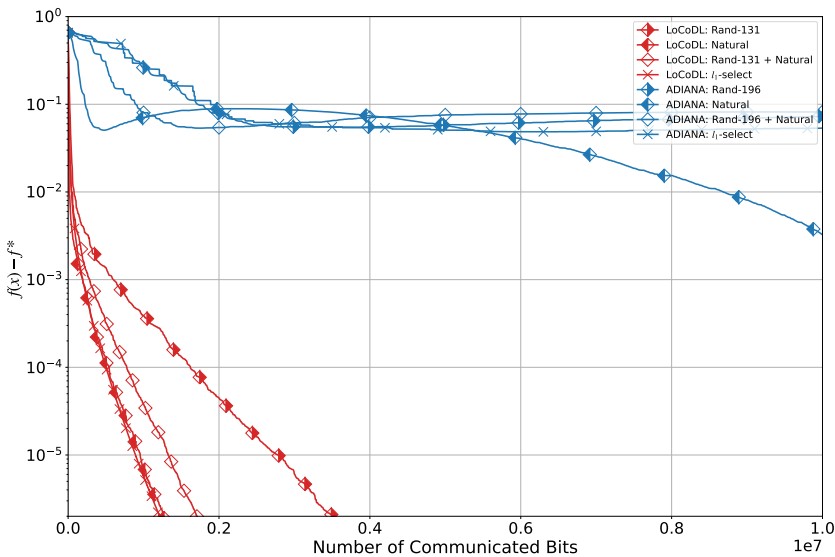

Figure 6: Comparison of LoCoDL and ADIANA using several compressors for logistic regression. The task was classifying 7s and 8s from the MNIST (LeCun et al., 1998) dataset, which consists of $d = 28 \times 28 = 784$ dimensions and 14,118 data points. We chose $n = 6$ (100× smaller than $d$).

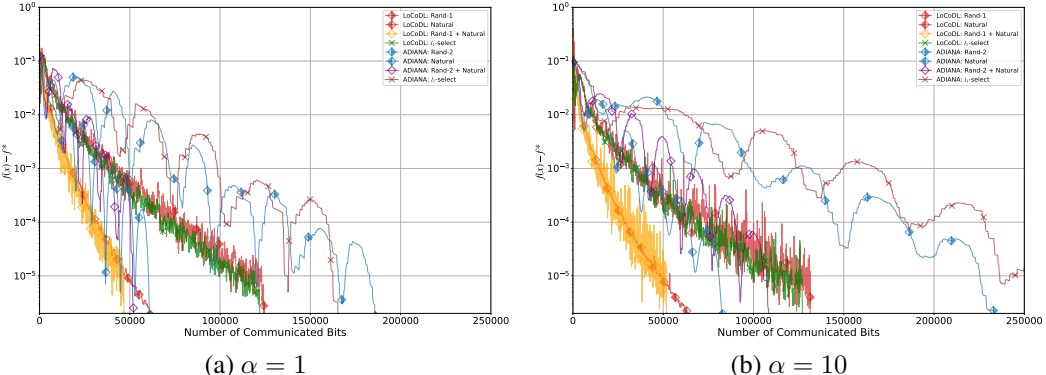

(a) $\alpha = 1$        (b) $\alpha = 10$

Figure 7: Comparison of LoCoDL and ADIANA with various compressors on logistic regression ($n = 100$, $d = 10$), with samples from the Dirichlet distribution of parameter $\alpha$, with $\alpha = 1$ on the left and $\alpha = 10$ on the right.

