# OpenReview forum: "LoCoDL: Communication-Efficient Distributed Learning with Local Training and Compression"
_ICLR.cc/2025/Conference — ICLR 2025 Spotlight_

### Official Review · Reviewer_ycJW · 2024-10-21

**Soundness:** 3
**Presentation:** 3
**Contribution:** 3
**Rating:** 6
**Confidence:** 3

**Summary:**

The paper combines local update with compression to improve communication efficiency in distributed estimation.

**Strengths:**

Proves the proposed algorithm the accelerate communication complexity due to both components used.

**Weaknesses:**

Only demonstrate strongly convex case. Assuming g is smooth (exclude non-smooth penalty such as lasso, for example)

**Questions:**

1. What is the intuition that make the proposed algorithm work compared to existing works that combines LT and CC?
2. Is the rate obtained optimal?

---

> ### Author Response · Authors · 2024-11-17
>
> Thank you for your positive evaluation.
>
> 1. What is the intuition that make the proposed algorithm work compared to existing works that combines LT and CC?
>
> LoCoDL is close in spirit to CompressedScaffnew. But CompressedScaffnew only works with one particular permutation-type compressor, that is linear and correlated. LoCoDL works with any unbiased compressors. This is obtained by constructing two, instead of one, model estimates $x$ and $y$, and compressing their difference, that tend to zero. The update of $y$ using $\nabla g$ is typically cheap so the increase in computation time is minor.
>
>
> 2. Is the rate obtained optimal?
>
> No, ADIANA has the optimal rate, as visible in Table 1, but the rate of LoCoDL and 5GCS-CC are not optimal, because of the $\sqrt{\omega}$ factor. However, our experiments show that ADIANA is slow in practice. So, ADIANA has the optimal asymptotic complexity, but it seems with a large constant hidden in the big-O notation. The paper on ADIANA of He et al. "Unbiased compression saves communication in distributed optimization: When and how much?", Neurips 2023, is recent and there is still a lot we don't understand on how to efficiently use Nesterov or other types of acceleration in stochastic algorithms. In the future, an even better algorithm than ADIANA with the same optimal complexity might be discovered. In the meantime, from the experiments we have conducted, LoCoDL is the new state of the art in practice.
>
> Weaknesses: Only demonstrate strongly convex case. Assuming $g$ is smooth (exclude non-smooth penalty such as lasso, for example)
>
> We believe we can show a $O(1/T)$ rate for the general convex case, with a similar analysis as in Condat et al. "Provably Doubly Accelerated Federated Learning: The First Theoretically Successful Combination of Local Training and Communication Compression" Appendix C. In our case there is the additional function $g$ so the analysis requires extra technicality.
>
> Adding a nonsmooth regularizer $R(x)$ in the problem, with its proximity operator applied at line 11 of LoCoDL (if $\rho=1$) might be possible, but this is far from obvious. With different nonsmooth functions $R_i$ at the clients, linear convergence can probably not be achieved, according to the discussion in Alghunaim et al. "Decentralized Proximal Gradient Algorithms With Linear Convergence Rates".

---

### Official Review · Reviewer_k7Jx · 2024-10-31

**Soundness:** 3
**Presentation:** 3
**Contribution:** 3
**Rating:** 8
**Confidence:** 3

**Summary:**

This paper proposes LoCoDL, an algorithm for distributed optimization that incorporates both local training and communication compression. The design of LoCoDL uses a primal-dual approach and randomization, provably achieving double acceleration in strongly convex landscape with unbiased compression. Empirical advantages of LoCoDL are demonstrated from experiments.

**Strengths:**

* The design of LoCoDL shows novelty such as the introduction of primal-dual updates.
* The convergence of LoCoDL with unbiased compression under strongly convex landscape is supported by rigorous theoretical analysis, provably showing a reduced communication overhead.
* LoCoDL achieves empirical advantages in experiments on regularized logistic regression.
* The presentation of the paper is well-organized and clear to follow.

**Weaknesses:**

* The theory of this paper has some limitation. For example, it assumes the usage of unbiased compression, which does not encompass some popular biased schemes such as Top-k compression. The proof also exploits strong convexity and nonconvex landscape poses a challenge.
* Empirical evaluations are conducted only on regularized logistic regression with homogeneous distribution of local dataset. The experiments are not able to demonstrate the performance in heterogeneous settings (which is claimed by the theory) and nonconvex problems; also see Questions.

**Questions:**

* Please consider a more comprehensive experimental evaluation.
  * Since nonconvex landscape is not covered by theory, experiments on deep neural networks can be done to show its performance in handling nonconvex problems.
  * The theory claims heterogeneous setting but experiments are done on a homogeneous distribution of dataset. Experiments with data heterogeneity can be conducted, as in previous related works like [1,2].
* I suggest the authors provide more discussions on the primal-dual design. How does this help with your proof? What challenges will you encounter without this primal-dual approach?

[1] Huang, X., Chen, Y., Yin, W., and Yuan, K. (2022). Lower bounds and nearly optimal algorithms in distributed learning with communication compression. Advances in Neural Information Processing Systems, 35, 18955-18969.

[2] Chen, S., Li, Z., and Chi, Y. (2024). Escaping Saddle Points in Heterogeneous Federated Learning via Distributed SGD with Communication Compression. In International Conference on Artificial Intelligence and Statistics (pp. 2701-2709). PMLR.

---

> ### Author Response · Authors · 2024-11-25
>
> Thank you for your positive evaluation and constructive comments.
>
> 1. Since nonconvex landscape is not covered by theory, experiments on deep neural networks can be done to show its performance in handling nonconvex problems.
>
> We are working in our team on many aspects of optimization, including nonconvex algorithms to train deep neural networks. But for this particular project, we did not run nonconvex experiments, because it is more on the theoretical side and our goal is to understand the mathematical and algorithmic principles that govern the achievable performance of local training and compression. So, we focus on the specific setting of strongly convex functions and unbiased compressors, and analyze the linear convergence rate. There is already a lot to study in this setting. In our experience, methods that work well in the convex setting, such as ADIANA, are not the ones that perform best for nonconvex training. Notably, the methods we consider are not the ones considered in the papers [1][2] you mention (thank you for pointing out these interesting references, we will include them in the final version.) So, there is a significant gap, and we believe this work is a step forward for a better understanding of compression in distributed training. But it is premature to apply LoCoDL to deep learning at this point. For instance, we would like to use biased compressors such as top-k in practice, but this requires a different error feedback technique and this becomes a different method. We can mention the paper Yi et al. "FedComLoc: Communication-Efficient Distributed Training of Sparse and Quantized Models", in which the empirical performance of algorithms using local training and compression is thoroughly investigated in practical deep learning scenarios. Again, there is still a lot to be done to bridge the gap between practically-efficient methods and theoretically-grounded algorithms like LoCoDL.
>
> 2. The theory claims heterogeneous setting but experiments are done on a homogeneous distribution of dataset. Experiments with data heterogeneity can be conducted, as in previous related works like [1,2].
>
> No, the datasets we use in the experiments are not homogeneous. Why do you think so? The number of data points is mot much larger than $n$, so there is no reason that the local distributions made by the data points assigned to the clients are similar in any way. For instance in Figure 3 (c) there are 690 datapoints and n=225, so 3 data points per client. In Figure 4 (b) there are 2477 data points and n=619, so 4 data points per client. These data points are different from each other.
>
> Our theory shows convergence and a convergence rate that are completely independent on the data heterogeneity. A finer analysis might be able to show that LoCoDL can exploit data homogeneity if there is any, with an even better rate, but we did not explore this direction. This is also related to personalized federated learning, yet another interesting topic.
>
> In any case, we have run additional experiments and revised the paper, with the added section B.1 in the Appendix. We have used samples from a Dirichlet distribution with different values of the parameter alpha, like in [2], to see whether the degree of heterogeneity influences convergence in this setting. LoCoDL again outperforms ADIANA, and for the different parameter values we tried, we did not observe a significant influence of the parameter alpha.
>
> 3. I suggest the authors provide more discussions on the primal-dual design. How does this help with your proof? What challenges will you encounter without this primal-dual approach?
>
> The primal-dual view was a source of inspiration for this work. First, in their paper about RandProx, Condat and Richtarik showed that control variates used to compensate the random errors due to probabilistic communication and compression can be viewed as dual variables in a primal-dual algorithm with randomized dual update. Second, there is a set of papers on decentralized optimization, with algorithms such as LEAD in Liu et al. "Linear convergent decentralized optimization with compression" ICLR 2021, where not 1 but 2 primal and dual variables are maintained at every node, and the differences between the 2 primal variables are compressed  (but there is no acceleration with respect to n and kappa in such papers. On the other hand, we currently don't know how to extend LoCoDL to the decentralized setting). This idea of doubling the primal and dual variables was the second source of inspiration for LoCoDL.

---

> > ### Comment · Reviewer_k7Jx · 2024-11-26
> >
> > Thank you for the detailed reply. My questions are sufficiently addressed and I am happy to raise my score from 6 to 8.

---

### Official Review · Reviewer_fCdQ · 2024-11-01

**Soundness:** 3
**Presentation:** 3
**Contribution:** 3
**Rating:** 8
**Confidence:** 4

**Summary:**

This paper studies how to develop communication-efficient algorithms for centralized optimization problem by combining the idea of local training and communication compression, and proposes an algorithm called LoCoDL. LoCoDL is both validated theoretically through a nearly optimal convergence rate and via empirically by comparing with previous SOTA algorithms.

**Strengths:**

1. Though the area of local training and communication compression has been explored by previous works, it may still be challenging to develop algorithms combining these ideas to achieve the theoretically optimal convergence rate. LoCoDL is shown to be optimal in cases where $n$ is smaller than the problem dimension $d$, which is often the case in real settings. Considering the assumptions used are standard, this result is believed to be strong enough.
2. It is impressive that LoCoDL performs much better than ADIANA, which has been the SOTA algorithm in distributed deterministic convex optimization with communication compression, and enjoys an optimal convergence rate matching the convergence lower bound.

**Weaknesses:**

1. The theoretical results are not perfect. Since LoCoDL has a sub-optimal rate in the case where $n$ is larger than $d$, its unclear whether it is the algorithm or the analysis is sub-optimal. As ADIANA has already achieves the optimal rate, it should be better if the author discuss the challenges to achieve a same rate.
2. It's not clear how LoCoDL is derived by combining the two ideas "local steps with local models" and "compressing the difference of two estimates". It is recommended that the authors discuss the motivation behind each algorithm line.

**Questions:**

1. It is a little bit strange that ADIANA performs worse than other baseline algorithms. Are the hyperparameters not sufficiently tuned?
2. Since with communication compression only we can achieve the optimal theoretical convergence rate, how to illustrate the theoretical benefit of using "local steps with local models"?

---

> ### Author Response · Authors · 2024-11-17
>
> Thank you for your positive evaluation, acknowledging the significance of our contributions.
>
> 1. Since LoCoDL has a sub-optimal rate in the case where $n$ is larger than $d$, it's unclear whether it is the algorithm or the analysis is sub-optimal. As ADIANA has already achieves the optimal rate, it should be better if the author discuss the challenges to achieve a same rate.
>
> We believe our analysis is tight and the combination of local training and compression cannot achieve the same rate as ADIANA, which is based on Nesterov-accelerated gradient steps. Indeed, when $n$ is large, in (7) we see that the primal term in the iteration complexity scales like $\kappa$ whereas the dual term scales like $(1+\omega)/p^2$. Therefore, to balance the 2 terms, we choose in (8) $p$ as $\sqrt{(1+\omega)/\kappa}$. At the end, the $\sqrt{\omega}$ factor is unavoidable, even if $n$ is huge.
> In ADIANA, the acceleration with respect to $\kappa$ is obtained directly in the primal term via momentum, and there is no need to balance the primal and dual errors, which yields the optimal $\tilde{O}(\sqrt{\kappa}+\omega)$ complexity. In other words, in ADIANA the compression error is decoupled from the acceleration, and we don't think it is possible to obtain this decoupling in local-training type algorithms. This is because of the $p^2$ dependence of the dual update: one $p$ comes from infrequent update, the second $p$ comes from the small dual stepsize to mitigate randomness, see the $p$ in lines 12 and 14 of LoCoDL.
>
> 2. It's not clear how LoCoDL is derived by combining the two ideas "local steps with local models" and "compressing the difference of two estimates". It is recommended that the authors discuss the motivation behind each algorithm line.
>
> We can add some text in Section 2.2 to highlight the ideas behind the operations. On one hand, for the algorithm to be variance-reduced, we need to compress vectors that tend to zero, that is why we compress differences between 2 model estimates $x$ and $y$. On the other hand, for acceleration to be possible, there must be progress toward the solution at every iteration, regardless of how rough the stochastic communication process is, that is why the estimates $\hat{x}_i^t$ and $\hat{y}^t$ are formed at every iteration $t$ via gradient descent steps with respect to the functions $f_i$ and $g$. Please let us know if there is a specific aspect you want us to explain.

---

> > ### Comment · Reviewer_fCdQ · 2024-11-26
> > **Thank you for your rebuttal**
> >
> > Thanks very much for the detailed response. My concerns are well addressed and I'll maintain my score.

---

### Official Review · Reviewer_1VVx · 2024-11-02

**Soundness:** 3
**Presentation:** 3
**Contribution:** 3
**Rating:** 8
**Confidence:** 3

**Summary:**

This paper proposed the LoCoDL algorithm, which applies both communication compression and local updates in distributed training, and analyzed its complexity, showing that with suitable compression scheme the total communication complexity matches that of the accelerated compressed methods

**Strengths:**

The proposed method, LoCoDL manages to combine both local updates and communication compression, and show that in the strongly convex regime, the method enjoys the communication-saving benefits of both techniques. LoCoDL combines ideas from ProxSkip and EF21 (and the likes) with primal-dual updates to achieve this impressive result.

**Weaknesses:**

see questions.

**Questions:**

1. Can the authors please explain in more details and provide some intuitions on why does LoCoDL "require" a regularisation $g$ to work? I put "require" in quotes because, as explained in section 3.1, one can reduce the case with $g=0$ to some strongly convex $\tilde g$. But such reduction highlights the fact that the algorithm, on its own, does need a strongly convex $g$, which seems quite mysterious to me. I think it would significantly improve my understanding of the technique if the authors can point out why/if LoCoDL, as is and without the reduction, would fail when $g=0$.

2. Based on the choice of the Lyapunov function, it seems to me that the convergence of LoCoDL is characterised in terms of $\frac{1}{n}\sum\lVert x_i^t-x^\star\rVert^2$. While ProxSkip's analysis also uses this convergence criteria, there the situation is slightly different since if the composite part can attain infinity then there is just no way to prove convergence in terms of the optimality gap when the prox step is skipped. But here the situation seems different, I wonder if it's possible to prove the convergence of LoCoDL in terms of the optimality gap, say, $f(\bar x) + g(\bar x)-f(x^\star)-g(x^\star)$ where $\bar x$ is the average of the primal variable (or perhaps the average optimality gap of each clients)? If not, what's the difficulty here? I think some of the algorithms that the authors compare LoCoDL against do use optimality gap as the convergence criteria, so there might be some mismatch.

3. The paper focuses on the strongly convex case. While the authors mentioned that non convex case might require significantly different techniques, I wonder what about the general convex case? It seems that LoCoDL would fail in that case (see also my question regarding $g$), but can the authors provide some insights in what would fail in the general convex case?

4. I wonder if LoCoDL can handle stochastic gradients? It seems to me that the idea of compressing the difference between two estimates works well with full gradient computations, but might not work so well when there is noise (e.g. EF21 fail to converge beyond the variance of the noise and ProxSkip's original analysis cannot achieve linear speedup with the noise). Is it also the case here? Have the authors tried the stochastic case?

---

> ### Author Response · Authors · 2024-11-17
>
> Thank you for your positive evaluation.
>
> 1. Can the authors please explain in more details and provide some intuitions on why does LoCoDL "require" a [strongly convex] regularisation g to work? [...This] seems quite mysterious to me.
>
> This is an excellent question and you are touching on the core of the approach. On one hand, for the algorithm to be variance-reduced, we need to compress vectors that tend to zero, that is why we compress differences between 2 model estimates $x$ and $y$. On the other hand, for acceleration to be possible, there must be progress toward the solution at every iteration, regardless of how rough the stochastic communication process is. So, at lines 4 and 5 of LoCoDL, we want the operators applied to $x^t$ and $y^t$ to be contractive. That is why both $f_i$ and $g$ have to be strongly convex.
>
> Without strong convexity of $g$, linear convergence is possible, but there is no acceleration (i.e. a dependence on $\kappa$ instead of $\sqrt{\kappa}$). This is because in this case, whenever communication occurs, we compress a difference between $x$ and $y$ but $y$ is outdated, it dates back to the last update/epoch and did not benefit from the local steps. This is similar to what happens in Scaffold, and this is why Scaffnew is superior to Scaffold.
>
> 2. I wonder if it's possible to prove the convergence of LoCoDL in terms of the optimality gap.
> 3. What about the general convex case?
>
> We did not do it but it is certainly possible to derive convergence results based on the objective gap, with sublinear convergence in the general convex case. In the paper "Provably Doubly Accelerated Federated Learning:
> The First Theoretically Successful Combination of Local Training and Communication Compression" Appendix C, Condat et al. derive a convergence analysis showing convergence of the Bregman distances $D_{f_i}(\bar{x}_i^t)$ and the consensus error $\sum_i \|\bar{x}_i^t-\bar{x}^t\|^2$, where the $\bar{x}_i^t$ are local averages over the iterates and $\bar{x}^t$ is their average. From these 2 properties, we can bound $f(\bar{x}^t)-f(x^\star)$. In our case there is $g$ in addition to $f$, but the analysis would be similar with added technicalities.
>
> 4. I wonder if LoCoDL can handle stochastic gradients?
>
> Yes, it is relatively easy to consider unbiased stochastic gradients with variance $\sigma^2$ and to prove linear convergence up to a neighborhood of size $O(\sigma^2)$. This has been done for Proxskip, see also the analysis in Condat et al. "TAMUNA: Doubly Accelerated Distributed Optimization with Local Training, Compression, and Partial Participation". There is no linear speedup with $n$, however. Getting linear speedup in local-training type algorithms is a difficult and long-standing question. There are some results in Guo et al. "Revisiting Decentralized ProxSkip: Achieving Linear Speedup". To the best of our knowledge, it is an open question whether we can get at the same time acceleration from local training and linear speedup with respect to independent stochastic gradients at the clients.
>
> Using variance-reduced stochastic gradient estimators, such as SAGA or SVRG, can certainly be done, following the ideas in Malinovsky et al. "Variance reduced ProxSkip: Algorithm, theory and application to federated learning." This makes the whole analysis more complicated, since the convergence analysis now depends on the properties of these estimators. Also, if we consider stochastic gradients of $g$, which is shared by all clients, the properties that need to be satisfied by such estimators are not obvious at first glance.

---

> > ### Comment · Reviewer_1VVx · 2024-11-21
> >
> > I would like to thank the authors for the explanation. I have no further questions and will raise my score.
> >
> > Please consider incorporating some of the discussions above into the revised version, especially on the role of strongly convex $g$ and the optimality gap (it would be even better if you can directly add the derivations).

---

> > > ### Author Response · Authors · 2024-11-25
> > >
> > > Thank you for your positive feedback and improving your score.
> > >
> > > We will follow your suggestions and add in the final version some explanations, and if time permits a convergence result based on the objective gap in the general convex case.
> > >
> > > In the meantime, we have added an experiment in Section B.1 in response to Reviewer k7Jx.

---

### Comment · Area_Chair_h31J · 2024-11-26
**Reviewers: Please go through the response**

Dear Reviewers,

The authors have provided their rebuttal to your questions/comments. It will be very helpful if you can take a look at their responses and provide any further comments/updated review.

Thanks!

---

### Meta-Review · Area_Chair_h31J · 2024-12-20

**Metareview:**

This paper studies communication efficient training methods in federated learning. Local training, and compression of gradients are two different methods for reducing communication in federated learning. This paper provides an algorithm that combines both and achieves convergence for strongly convex objective functions. As such, it is interesting to know that this works.

As far as weaknesses are concerned, this is not a very rigorous paper as far as compression parts are concerned (terms such as "communication complexity in reals" shows the lack of understanding of issues in communication, which is a pretty big engineering field by itself). However, from optimization perspective (federated learning) this paper has some values, and the reviewers do recognize that. From the tables of the papers, there is no theoretical advantage this paper provides over existing ones, but in simulation it is better.

Overall, I recommend acceptance.

**Additional Comments On Reviewer Discussion:**

Reviewers are generally positive and increased their scores after author response.

---

### Decision · Program_Chairs · 2025-01-22

Accept (Spotlight)